# Atomically dispersed asymmetric cobalt electrocatalyst for efficient hydrogen peroxide production in neutral media

Longxiang Liu[1,8], Liqun Kang [2,8], Jianrui Feng[1,8], David G. Hopkinson [3], Christopher S. Allen[3,4], Yeshu Tan[1], Hao Gu[5], Iuliia Mikulska [6], Veronica Celorrio [6], Diego Gianolio [6], Tianlei Wang[1], Liquan Zhang[1], Kaiqi Li[1], Jichao Zhang[1], Jiexin Zhu[1], Georg Held [6], Pilar Ferrer [6], David Grinter [6], June Callison [7], Martin Wilding [7], Sining Chen[1], Ivan Parkin [1] ✉ & Guanjie He [1] ✉

Electrochemical hydrogen peroxide ($H_2O_2$) production (EHPP) via a two-electron oxygen reduction reaction ($2e^-$ ORR) provides a promising alternative to replace the energy-intensive anthraquinone process. M-N-C electro-catalysts, which consist of atomically dispersed transition metals and nitrogen-doped carbon, have demonstrated considerable EHPP efficiency. However, their full potential, particularly regarding the correlation between structural configurations and performances in neutral media, remains underexplored. Herein, a series of ultralow metal-loading M-N-C electrocatalysts are synthesized and investigated for the EHPP process in the neutral electrolyte. CoNCB material with the asymmetric Co-C/N/O configuration exhibits the highest EHPP activity and selectivity among various as-prepared M-N-C electrocatalyst, with an outstanding mass activity ($6.1 \times 10^5 A g_{Co}^{-1}$ at 0.5 V vs. RHE), and a high practical $H_2O_2$ production rate (4.72 mol $g_{catalyst}^{-1} h^{-1} cm^{-2}$). Compared with the popularly recognized square-planar symmetric Co-$N_4$ configuration, the superiority of asymmetric Co-C/N/O configurations is elucidated by X-ray absorption fine structure spectroscopy analysis and computational studies.

Due to versatile applications and end-use industries, hydrogen peroxide ($H_2O_2$) demand has witnessed rapid growth in recent decades (Fig. 1a). The global hydrogen peroxide market is expected to reach USD 4.0 billion by 2027 with an annual growth rate of 5.0%[1]. Currently, industrial $H_2O_2$ production is highly dependent on the anthraquinone process to produce high concentrations of up to 70%[2]. However, this process involves energy-intensive multiple steps, including complex hydrogenation, oxidation, extraction, and post-treatment, leading to extra waste generation and environmental concerns[3]. Moreover, depending on the end-use industries, $H_2O_2$ solutions with low concentrations (<6%) are usually more favorable for many practical applications, such as agricultural irrigation and disinfection[4]. For example, a very diluted $H_2O_2$ solution with a concentration of 0.1% exhibits excellent bacteria-killing performance in water disinfection

[1]Christopher Ingold Laboratory, Department of Chemistry, University College London, London WC1H 0AJ, UK. [2]Department of Inorganic Spectroscopy, Max-Planck-Institute for Chemical Energy Conversion, Stiftstr. 34-36, 45470 Mülheim an der Ruhr, Germany. [3]Electron Physical Science Imaging Centre, Diamond Light Source, Harwell Science and Innovation Campus, Didcot OX11 0DE, UK. [4]Department of Materials, University of Oxford, Parks Road, Oxford OX1 3PH, UK. [5]Department of Chemical Engineering, University College London, London WC1E 7JE, UK. [6]Diamond Light Source, Harwell Science and Innovation Campus, Didcot OX11 0DE, UK. [7]UK Catalysis Hub, Research Complex at Harwell, Rutherford Appleton Laboratory, Didcot OX11 0FA, UK. [8]These authors contributed equally: Longxiang Liu, Liqun Kang, Jianrui Feng. ✉e-mail: i.p.parkin@ucl.ac.uk; g.he@ucl.ac.uk

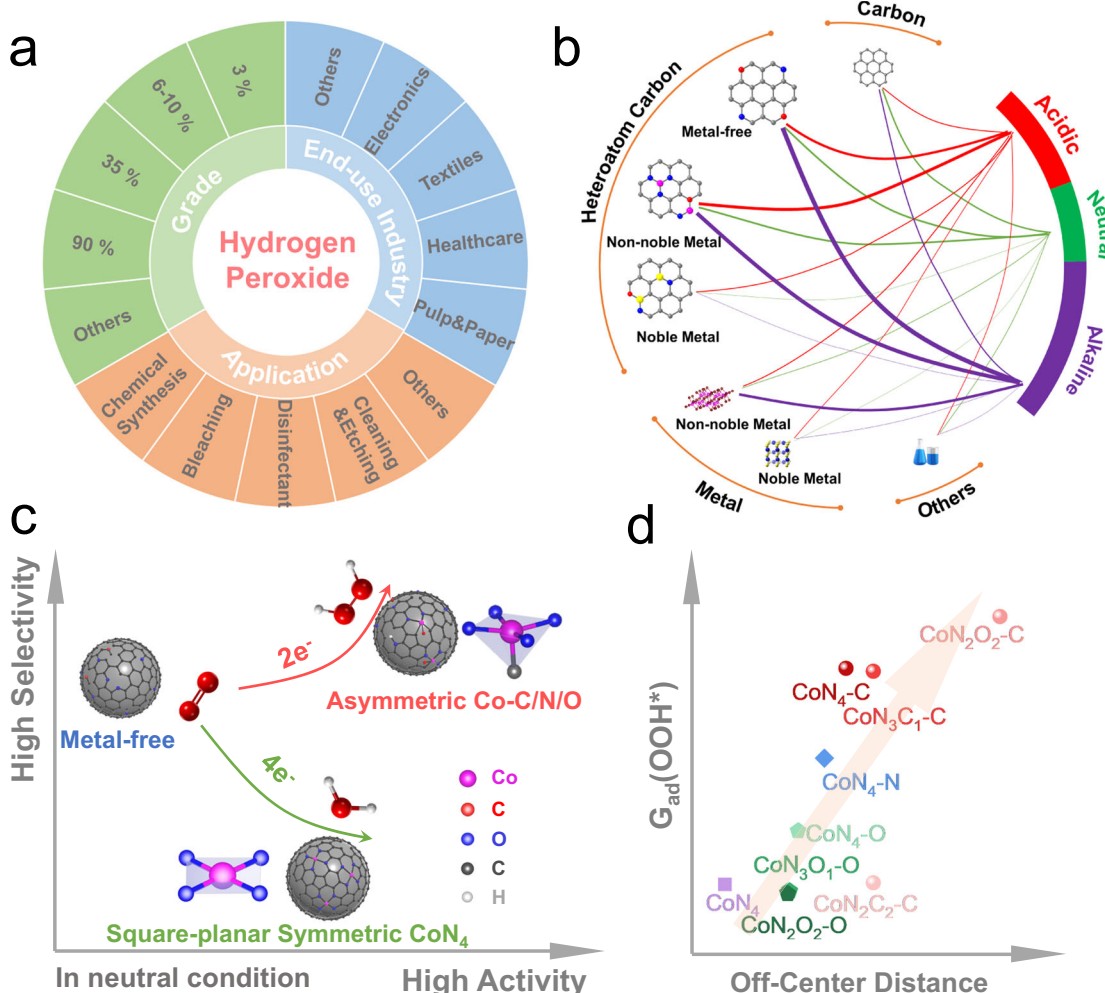

**Fig. 1 | Schematic illustration of asymmetric Co-C/N/O configurations. a** The applications, end-use industries, and grades of the hydrogen peroxide market. **b** The concept map showing the prevalence of EHPP electrocatalysts by pH of electrolytes. The classification details are provided in Table S1–2. **c** The schematic illustration of the EHPP selectivity and activity of centrosymmetric CoN₄ and asymmetric Co-C/N/O electrocatalysts in 0.1 M PBS (pH = 7). **d** The relationship between the off-center distance of Co atom and the adsorption energy of OOH*.

treatment[5]. As a promising alternative, electrochemical $H_2O_2$ production (EHPP) via a two-electron oxygen reduction reaction (2e⁻ ORR) provides a feasible and green approach for on-site $H_2O_2$ production. This strategy could circumvent the issues with the anthraquinone process and greatly reduce the cost of storage and transportation of $H_2O_2$ products, thus attracting growing interest (Fig. S1).

Electrocatalysts with high 2e⁻ ORR selectivity and activity are highly desirable to improve the efficiency of EHPP. Depending on electrolyte pH, different types of electrocatalysts have been developed, including carbon materials, heteroatom-doped carbon materials, non-noble metal materials, and noble metal materials. As displayed in Fig. 1b and Table S1, 2, previous literature analysis reveals that the EHPP study under alkaline conditions is most popular for various electrocatalysts, which is due to the superior activities of electrocatalysts compared to those under acidic and neutral conditions[6–8]. The popularity of EHPP study under acidic conditions primarily stems from the application of the electro-Fenton process to produce hydroxyl radicals (•OH) for the degradation of aqueous organic pollutants, and non-noble metal heteroatom-doped carbon electrocatalysts have been extensively researched in this context[8,9]. In contrast to the extensive research of EHPP under alkaline and acid conditions, less efforts have been devoted to the 2e⁻ ORR study in neutral media due to the low concentration of H⁺ and OH⁻ that

negatively affects the ORR kinetics[10,11]. However, the utilization of neutral $H_2O_2$ solution is more popular and versatile for practical applications, such as household cleaning and gardening, in part as it eliminates the potential secondary cost associated with neutralizing acid and alkaline solutions[8,12]. Therefore, it is highly desirable to promote the development of electrocatalysts for EHPP in neutral media, including the design of efficient electrocatalysts, study of novel mechanisms, and endeavor in practical productions.

M-N-C electrocatalysts with atomically dispersed transition metals (TMs) and nitrogen species anchored on the carbon support have been extensively investigated for diverse electrocatalytic reaction studies[13]. The merits of exceptional activity, superior selectivity, and extremely high utilization efficiency have aroused considerable interest for both 4e⁻ and 2e⁻ ORR studies in alkaline and acidic media[14–16]. To prepare M-N-C electrocatalysts, the annealing strategy is one of the popularly employed approaches, which incorporates the annealing of various carbon, nitrogen, and metal precursors[17]. The availability of multiple precursors bring out promising opportunities to study the structure-property-performance relationship and scrutinize profitable routes. Nevertheless, a number of challenges have delayed the development of M-N-C electrocatalysts derived from the annealing strategy. For example, the synthesis of M-N-C electrocatalysts with high-loading of 3d TMs (>5 wt%) is generally pursued to achieve the high activity[18–20].

Unfortunately, the fact that the majority of TM species are entombed into the bulk rather than at the surface of the carbon support, limiting active sites utilization, increasing precursor costs, and potentially deteriorating the intrinsic activity. Furthermore, considering the soaring prices and potential supply risks of some 3d TMs, optimized techniques and strategies are required to achieve the better utilization of active metal species[21]. Additionally, the ambiguous electronic configuration and structural disorder of M-N-C species in the annealed electrocatalyst pose another major challenge for accurately identifying the active sites, thus deepening the difficulty for precisely manipulating the electronic structures. The symmetrical square-planar M-N$_4$ models have been popularly proposed using synchrotron-based X-ray absorption fine structure (XAFS) spectroscopy analysis, including interpreting X-ray absorption near-edge structure (XANES) and fitting Fourier-transformed extended X-ray absorption fine structure (FT-EXAFS)[15,20,22–24]. However, the limitations of fitting FT-EXAFS have been sometimes arbitrarily neglected in distinguishing M-C/N/O coordination with similar bond lengths due to close atomic number, thus leading to ambiguous or even somewhat misleading[25]. Therefore, deeper insights are needed to understand the electronic structures, such as clarification of coordination environment of M-N-C catalysts, with the aim of revealing the underlying structure-performance relationships[26,27].

In this work, a surface engineering strategy was employed to anchor different 3d TMs (Mn, Fe, Co, Ni, Cu) with ultralow loadings (<0.1 wt%) on the surface of N-doped carbon black. The surface-anchored M-N-C method greatly enhanced the exposure of active sites. The evaluation of their EHPP performance in neutral electrolytes was carried out and the CoNCB material with asymmetric Co-C/N/O configuration showed the highest activity and selectivity among various M-N-C materials. The H$_2$O$_2$ selectivity of CoNCB is over 95% in a large potential range of 0.45 – 0.75 V, and the highest cobalt activity ($6.1 \times 10^5$ A/g$_{Co}$) and turnover frequency (186 s$^{-1}$) were achieved at 0.5 V vs. reversible hydrogen electrode (RHE). Furthermore, the atomically dispersed Co species, as well as the electronic structure was comprehensively revealed by high-angle annular dark-field scanning transmission electron microscopy (HAADF-STEM) and XAFS spectroscopy analysis. Multiple asymmetric Co-C/N/O configurations were proposed based on the XANES and FT-EXAFS analysis, and their superiority for EHPP performance was demonstrated by comparing with that of known Co-N$_4$ configuration (Fig. 1c), as well as computational validations. Furthermore, the computational study revealed the positive and quantitative correlation between the asymmetry of Co-C/N/O configuration and the adsorption energy of the OOH* intermediate, i.e., the superior adsorption energy of OOH* intermediate is dependent on the off-center distance of Co atom (Fig. 1d).

## Results

### Materials synthesis and characterization

The structural morphologies at nano and sub-nano scale levels were characterized by aberration-corrected scanning transmission electron microscopy in annular bright-field mode and high-angle annular dark-field mode (ABF-STEM and HAADF-STEM) in Fig. 2a–c. The compact lattice fringes of carbon black were preserved, and no nanoparticles can be observed in the ABF-STEM image (Fig. 2a) and low-magnification STEM images (Fig. S2). Due to the different Z-contrast, isolated bright spots of Co atoms were displayed in the HAADF-STEM image. In addition, Co atoms were primarily anchored on the distorted surface of carbon black support due to the protection of the compact lattice fringes, thus exposing most active sites and enhancing atomic utilization efficiency. Similarly, the atomic dispersion of Mn, Fe, Ni, and Cu was shown in Fig. S3. Of note, the HAADF-STEM signals are also dependent on the density and thickness of specimen[28]. The ABF- and HAADF-STEM images of metal-free NCB material were displayed in Fig. S4 for clarification of atomic dispersion of doped 3d transition

metals. The mass loading amounts of Mn, Fe, Co, Ni, and Cu were 0.04 wt%, 0.05 wt%, 0.05 wt%, 0.08 wt%, and 0.06 wt%, respectively, determined by MP-AES measurement.

To investigate the electronic structure of CoNCB, synchrotron radiation-based near-edge X-ray absorption fine structure (NEXAFS) measurement was first carried out including Co L-edge, N K-edge and O K-edge NEXAFS spectra. The high surface sensitivity of NEXAFS measurement in total electron yield (TEY) mode (<10 nm) makes it suitable to unravel the electronic structure of components at electrocatalysts' surfaces[29,30]. Co L-edge NEXAFS spectra derived from the dipole-allowed Co 2p to unoccupied 3d orbitals electronic transition (L$_3$: $2p_{3/2} \rightarrow 3d$, L$_2$: $2p_{1/2} \rightarrow 3d$) are shown in Fig. S5a. Cobalt porphyrin (CoPr) reference material with a low spin state displayed three characteristic peaks at around 777.9 eV, 779.5 eV, and 780.8 eV in the Co L$_3$-edge NEXAFS spectrum, assigned to the half-occupied $3d_{z^2}^1$ and unoccupied $3d_{x^2-y^2}^0$ orbitals due to the 3d orbital splitting of D$_{4h}$ symmetry (Fig. S6)[31–33]. Compared with the spectrum of the CoPr reference, the Co L$_3$-edge spectrum of CoNCB material showed two broadened peaks at around 777.5–778.9 eV and 779.5–781.5 eV, respectively. This probably indicated that the square-planar symmetric Co-N$_4$ configuration was absent in CoNCB material. In addition, because of the ultra-low loading of Co, the poor signal-to-noise ratio made it difficult to determine the accurate electronic orbitals.

Meanwhile, the electronic structure of oxygen and nitrogen functional groups on the surface of CoNCB was also evaluated in Fig. S5b. The O K-edge NEXAFS spectra displayed 6 characteristic peaks at 530.4 eV, 531.7 eV, 532.7 eV, 535.7 eV, 540.2 eV, and 546.1 eV, corresponding to O 1s transitions to unoccupied π* (C = O), π* (O-C = O), π* (C-O-C)/(C-OH), σ* (O-H), σ* (C-O), and σ* (C = O), respectively[34–36]. Compared with oxidized O-CB, the decreased peak intensity involving π* (C = O), π* (O-C = O), π* (C-O-C)/(C-OH) of CoNCB indicated the removal of ketone, carboxyl, and hydroxyl groups during the annealing process. Nevertheless, residual oxygen groups remained on the surface of CoNCB. The identification of N K-edge features was struggling due to the contamination in the measurement. A supplementary note was provided in the supporting information (Fig. S5c). The complex co-existence of oxygen and nitrogen functional groups heavily increased the difficulty of identifying the possible Co coordination environments in the following EXAFS analysis.

Furthermore, Co K-edge XAFS measurements were performed to reveal the electronic and geometric structure and the coordination environment of Co species in CoNCB material at the bulk level. The local electronic and geometric structures were elucidated by Co K-edge XANES analysis. As shown in Fig. 2d, CoPr reference exhibited a strong shoulder peak at 7715.2 eV in the rising edge, which was ascribed to typical 1s to 4p$_z$ and ligand-to-metal charge transfer configuration (LMCT) shakedown transition in the square-planar and centrosymmetric macrocyclic Co complexes (D$_{4h}$ symmetry) structures (Fig. S6)[37,38]. In addition, a very weak pre-edge was displayed at 7709.5 eV due to the dipole forbidden 1s to 3d electron transition[39]. In comparison with CoPr, the fingerprint of square-planar symmetric Co-N$_4$ structure disappeared in the rising edge of CoNCB and the pre-edge exhibited the peak incline along with the blue shift due to the increase in 3d and 4p orbital mixing via d-p hybridization (Fig. 2d inset image and Fig. 2e). This indicated the absence of the square-planar symmetric environment in the geometric structure of CoNCB, which might be due to the distortion effect by the axial coordination[37]. In addition, as displayed in the STEM images, the unique and distorted surface of carbon black support probably contributed to the formation of asymmetric configurations. We would like to mention that the position of the rising edge of reference materials cannot be referred to interpret the oxidation state of CoNCB material in this case due to the incomparable geometric

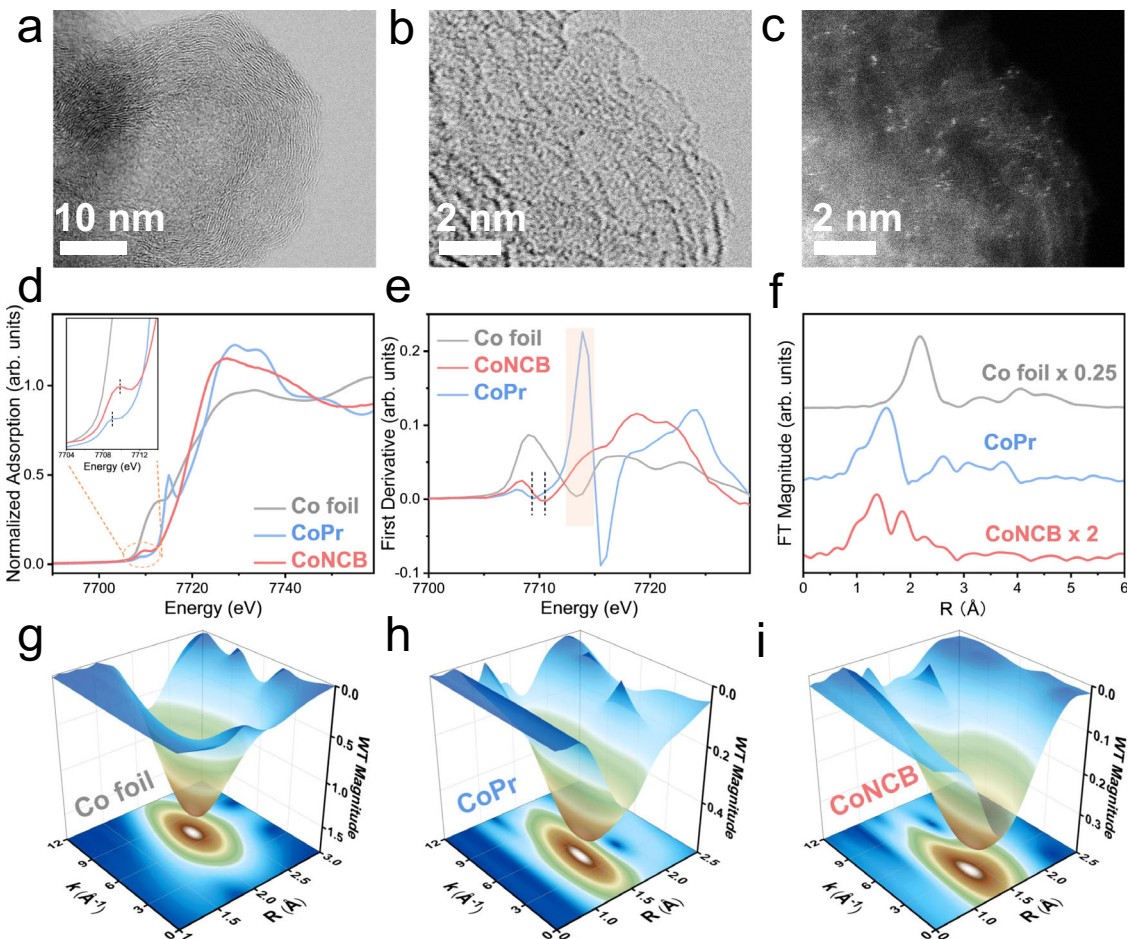

**Fig. 2 | Morphological and structural characterization. a, b** ABF-STEM and (**c**) HAADF-STEM images of CoNCB material. **d** Co K-edge XANES spectra. **e** First derivative of Co K-edge XANES spectra. **f** FT magnitudes of $k^2$-weighted EXAFS spectra without phase correction. Spectra are displaced vertically for clarity. $k^2$-weighted WT-EXAFS 3D contour plots of (**g**) Co foil, (**h**) CoPr, and (**i**) CoNCB.

structure between CoPr and CoNCB, as well as different metal-ligand bond covalencies[37].

The coordination environment of Co species was further investigated by Co K-edge EXAFS analysis, including Fourier-transformed EXAFS (FT-EXAFS) and Wavelet-transformed EXAFS (WT-EXAFS) analysis. Fitting of FT-EXAFS spectra provides the radial distance of backscattering atoms from the absorbing atom center[25]. In Fig. 2f, Co foil reference exhibited a major peak at 2.2 Å, and CoPr reference exhibited a major peak at 1.6 Å, as well as multiple scattering features in the range of 2.5−4 Å due to the square-planar centrosymmetric Co-N-C structure. In contrast, multiple scattering features disappeared in the spectrum of CoNCB, and a major peak at 1.4 Å along with a shoulder peak at 1.9 Å was observed in the spectrum of CoNCB. In addition, WT-EXAFS spectra are provided to provide the R-space and k-space at the same time. In the R-$k$ contour map (Fig. 2g–i), Co foil reference showed the maximum intensity at $k = 7.1$ Å$^{-1}$ and R = 2.2 Å, corresponding to Co-Co contribution. Due to the low atomic number of C/N/O, CoPr reference material exhibited the maximum intensity at $k = 4.6$ Å$^{-1}$ and R = 1.5 Å corresponding to the Co-N contribution. As for the CoNCB catalyst, the maximum intensity appeared at $k = 3.1$ Å$^{-1}$ and R = 1.4 Å, which is far away of the intensity of Co-Co scattering at $k = 7.1$ Å$^{-1}$, indicating the Co-C/N/O contributions. Of note, WT-EXAFS spectrum is not feasible to discern the contribution of individual Co-C, Co-N, or Co-O coordination due to the approximate $k$-space dependencies[40,41]. Nevertheless, the isolated presence of Co species in CoNCB catalyst was confirmed by FT-EXAFS and WT-EXAFS analysis, which is a good complement for the HAADF-STEM observation. Meanwhile, the isolated dispersions of Mn, Fe, Ni, and Cu at the bulk level were illustrated by FT-EXAFS and WT-EXAFS analysis (Fig. S7–10).

Based on the above EXAFS and NEXAFS analysis, the attempt to reveal the coordination atoms, number, and distance of Co species in the CoNCB catalyst was performed by fitting the Co K-edge FT-EXAFS spectra. Unfortunately, due to the complicated presence of C, N, and O elements and their close atomic number, it was found impossible to determine the accurate coordination environment of Co-C/N/O species. As displayed in Fig. S11–14 and Table S4, many possibilities existed depending on the used fitting paths, such as Co-N$_3$/O$_2$, Co-N$^1_3$/N$^2_2$, Co-C$_3$/N$_2$ configurations with five different coordination paths. Besides, the fact that EXAFS spectra are collected based on the average bulk information, and the systematic error exists in determining the coordination number, further deepens the struggle to identify the accurate coordination environment of Co species in CoNCB[25,42]. The same issues also existed in the EXAFS fitting of MnNCB, FeNCB, NiNCB, and CuNCB, which greatly increased the complexity. This differed from most of previous studies in which only one configuration was considered to perform the EXAFS fitting[15,22,43–45].

**EHPP performance**

The EHPP performance of as-obtained materials was evaluated using the rotating ring-disk electrode (RRDE) system in the O$_2$-saturated 0.1 M phosphate buffer solution (PBS, pH = 7.0). Due to the ultra-low loading of metal contents, we assumed that different transition metals played

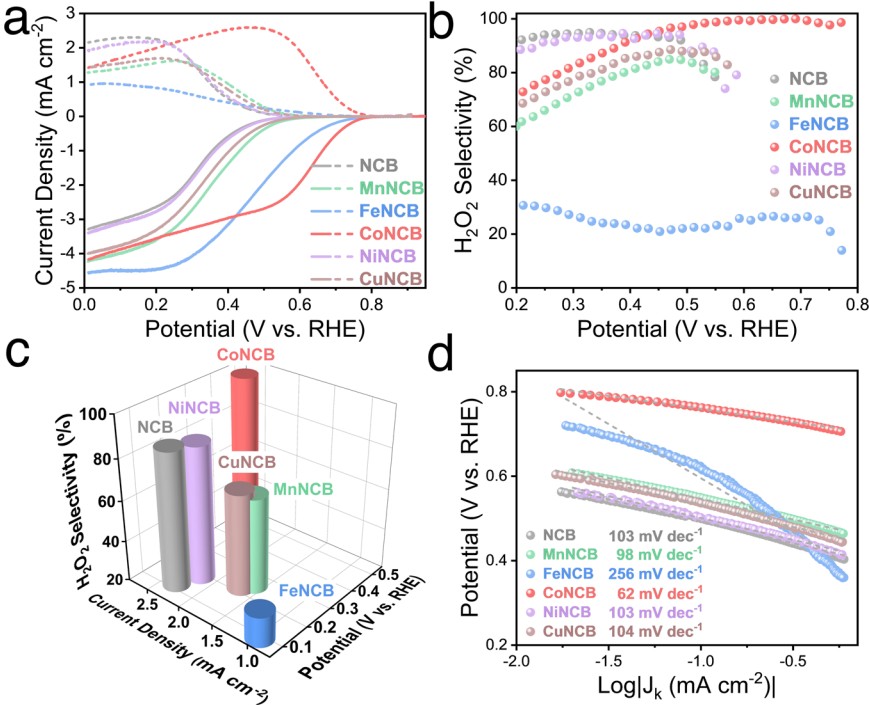

**Fig. 3 | EHPP activity and selectivity of CoNCB material. a** ORR polarization curves of disk current density (solid line) and ring current density (dash line) of NCB, MnNCB, FeNCB, CoNCB, NiNCB, and CuNCB in 0.1 M PBS (pH = 7). **b** Calculated $H_2O_2$ selectivity ($H_2O_2$ %). **c** Comparison of $H_2O_2$ selectivity at the potential of the maximum ring current density. **d** Calculated Tafel plots; labels show the corresponding Tafel slopes.

negligible roles in adjusting the structural morphologies of carbon black support, and the difference in EHPP performance was dominated by the different M-C/N/O species. As displayed in Fig. 3a, the onset potential (potential at 0.1 mA cm$^{-2}$) was more positive after introducing M-C/N/O species, which was in the order of CoNCB (0.76 V) > FeNCB (0.61 V) > MnNCB (0.55 V) > CuNCB (0.54 V) > NiNCB = NCB (0.50 V). In terms of the $H_2O_2$ selectivity, it was deteriorated by Fe-C/N/O, Mn-C/N/O, and Cu-C/N/O species. Regarding the maximum ring current density, the metal-free NCB reached 2.32 mA cm$^{-2}$ at 0.19 V, but it declined in the order of FeNCB (0.97 mA cm$^{-2}$ at 0.09 V) < MnNCB (1.62 mA cm$^{-2}$ at 0.25 V) < CuNCB (1.71 mA cm$^{-2}$ at 0.22 V), whereas it increased to 2.61 mA cm$^{-2}$ at 0.49 V for CoNCB (Fig. 3c). The kinetic activity was further evaluated through Tafel slope analysis, as displayed in Fig. 3d. The Tafel slopes were in the order of CoNCB (62 mV dec$^{-1}$) < MnNCB (98 mV dec$^{-1}$) < NCB = NiNCB (103 mV dec$^{-1}$) < CuNCB (104 mV dec$^{-1}$). In addition, the Tafel slopes determined by the disk LSV current density displayed similar behaviors (Fig. S15), exhibiting the fast kinetic activity for CoNCB material, compared to other prepared M-N-C electrocatalysts. The Tafel slop of CoNCB is far away from 120 mV dec$^{-1}$, probably indicating that the first charge transfer ($O_2$ + * + ($H^+$ + $e^-$) → OOH*) is not the rate-determining step[46]. Of note, the analysis of Tafel slopes of other M-N-C materials exhibited discrepancies with literatures[3,45], which might be due to the different metal loadings in the electrocatalysts.

Through performing performance contrast and kinetic analysis, we could conclude that Mn-C/N/O, Fe-C/N/O, and Cu-C/N/O species deteriorated the EHPP performance, whereas the Co-C/N/O species improved the EHPP activity and selectivity, compared with that of metal-free NCB materials. In addition, Ni-C/N/O species played a negligible role in adjusting EHPP activity and selectivity, which differed from previous reports[47,48]. Remarkably, besides the much-improved onset potential of 260 mV, CoNCB exhibited high $H_2O_2$ selectivity of over 95% in the wide range of 0.45–0.75 V, determining the significant role of Co-C/N/O active sites. The onset potential and $H_2O_2$ selectivity outperformed most of state-of-the-art EHPP electrocatalysts,

demonstrating its promising potential (Fig. 4a). In particular, due to the ultra-low loading of Co species, the Co activity reached over $6.1 \times 10^5$ A g$_{Co}^{-1}$ (at 0.5 V), which was over 500 times higher than that of reported Co-N-C electrocatalysts for 2e$^-$ ORR. The turnover frequency (TOF) values at 0.5 V of CoNCB and other reported Co-N-C based ORR electrocatalysts were displayed in Fig. 4c to reveal the intrinsic electrocatalytic activity. The TOF value of CoNCB reached 186 s$^{-1}$, which was 16 times higher than the currently reported Co-N-C catalyst, demonstrating the superior atomic electrocatalytic activity of CoNCB[24].

The effect of cobalt loading and catalyst loading was also evaluated, as shown in Figs. S16, 17. The optimal Co loading content was determined to be 0.05 wt% with both high activity and selectivity. Moreover, the EHPP stability of CoNCB was evaluated by the accelerated durability test (ADT) as shown in Fig. S18. After 5000 cycles of CV test, the potentials to reach 0.1 mA cm$^{-2}$ and 1 mA cm$^{-2}$ slightly declined by 20 mV and 39 mV after ADT, respectively, but the high $H_2O_2$ selectivity was maintained. Moreover, the practical EHPP was further evaluated using the flow cell, as shown in Fig. 4d–f and Figs. S19, 20. CoNCB material was uniformly coated on the gas diffusion electrode through the air-brush method with a loading of 0.2 mg cm$^{-2}$. The chronopotentiometry method was used at a constant current density of 20 mA cm$^{-2}$ for 5 h in 1 M PBS electrolyte. Figure 4e shows that initial CoNCB material exhibited more positive onset potential and larger current density than metal-free NCB material. The slight decrease in the LSV curve of spent CoNCB after 5 h was probably due to the partial leaching of electrolyte (Fig. S19). The cerium sulfate titration method was used to determine the generated $H_2O_2$ combined with UV-vis spectroscopy (Fig. S20). The average production rate reached 4.72 mol g$_{catalyst}^{-1}$ h$^{-1}$ cm$^{-2}$ with the Faradic efficiency (FE) of over 60%. The decline of FE was probably due to the slight leaching of electrolyte on the gas diffusion electrode (Fig. S19). Moreover, a stable potential with time was observed (Fig. S20d), further indicating the stability of CoNCB material for practical application.

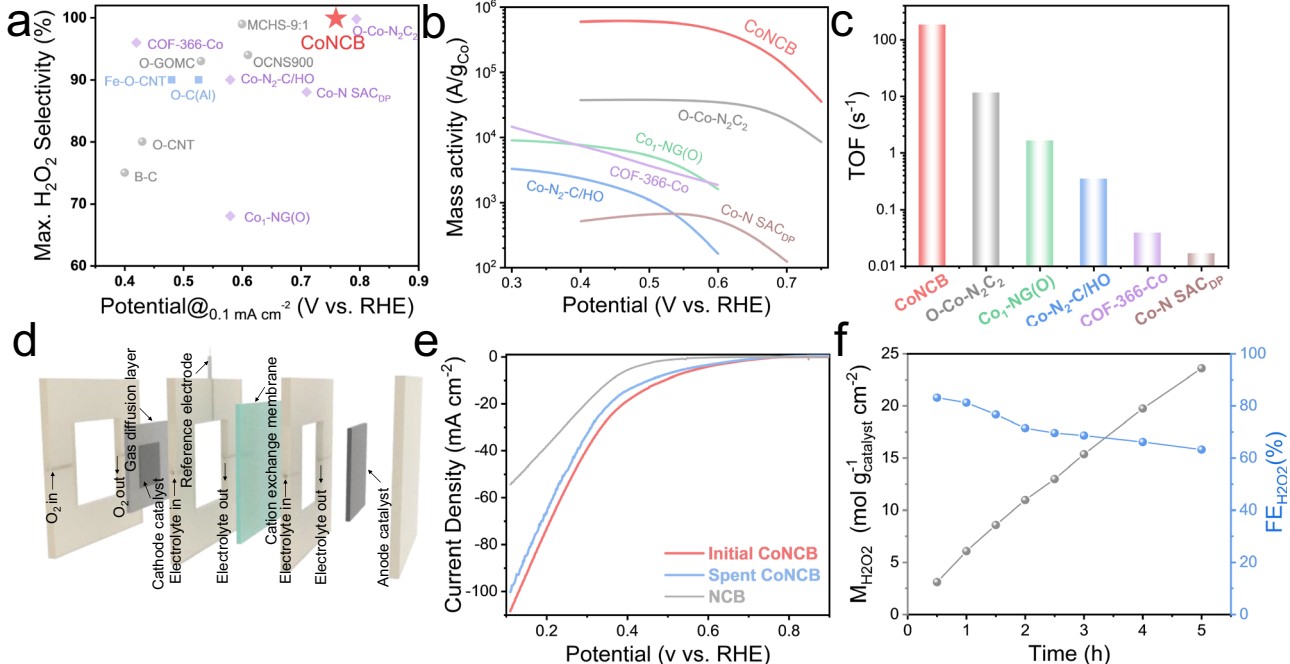

**Fig. 4 | Electrochemical flow cell performance. a** Onset potential and maximum $H_2O_2$ selectivity contrast in neutral electrolytes (gray circles: metal-free carbon electrocatalysts; purple diamonds: Co-N doped electrocatalysts; blue squares: other metal doped electrocatalysts). **b** The mass activity of the cobalt in Co-N doped electrocatalysts at 0.5 V. **c** TOF values at 0.5 V. **d** Schematic illustration of flow cell configuration for $H_2O_2$ production. **e** Comparison of LSV curves of NCB, initial CoNCB, and spent CoNCB after 5 h running. Electrolyte: 1 M PBS. **f** $H_2O_2$ production under 20 mA cm$^{-2}$ and related Faradic efficiency (FE).

The Co K-edge XAFS spectroscopy analysis of CoNCB material demonstrated an asymmetric geometric structure for the Co-C/N/O species. Herein, CoPr was deposited on blank CB to simulate the square-planar $D_{4h}$ symmetric Co-N$_4$ configuration, which was used as the contrast material to reveal the difference between the symmetrical Co-N$_4$ configuration and the asymmetric Co-C/N/O species. The EHPP performance in the neutral electrolyte of NCB, Co-N$_4$, and CoNCB was displayed in Fig. 5a. Compared with NCB, Co-N$_4$ also promoted the onset potential, confirming an improved EHPP activity by Co-N species. Nevertheless, the $H_2O_2$ selectivity of Co-N$_4$ was much lower than that of CoNCB (Fig. 5b). Moreover, the promoted $H_2O_2$ selectivity of CoNCB was also observed in acidic (pH = 1) and alkaline (pH = 13) electrolytes (Fig. S21). Figure 5c displays the comparison of the potential at 1 mA cm$^{-2}$ and the highest ring current density of Co-N$_4$ material and asymmetrical CoNCB electrocatalysts. We hypothesized that asymmetric Co-C/N/O configuration possesses superior binding energy of OOH* intermediate than symmetric Co-N$_4$ in the neutral environment, leading to a highly active and selective EHPP process.

While it was difficult to reveal the exact electronic structures of Co-C/N/O species in CoNCB material from the experimental characterization, density functional theory (DFT) computational study was performed to evaluate the underlying origin of its superior EHPP performance compared with that of square-planar symmetric Co-N$_4$ configuration. Herein, eight $C_{4V}$-alike asymmetric configurations with five Co-C/N/O coordination paths were proposed based on the above comprehensive analysis of XANES, FT-EXAFS, and NEXAFS analysis (Fig. 5d and Fig. S22), as well as the fact the surface was distorted in the CoNCB material. Their XANES spectra were simulated using Finite Difference Method Near Edge Structure (FDMNES) program, as shown in Fig. 5e. The simulated XANES spectra of Co foil and CoPr reference aligned well with the experimental spectra (Fig. S23). For these asymmetric Co-C/N/O configurations, the feature of dipole forbidden 1s to 3d electron transition at the pre-edge position

inclined due to the increasing 3d and 4p orbital hybridization. Besides, the fingerprint of square-planar symmetry at the shoulder peak position disappeared for all asymmetric configurations. The adsorption energy of OOH* intermediate ($G_{ad}(OOH^*)$) was used as the critical descriptor to determine the electrocatalytic activity of different configurations[5,49]. The calculated limiting potential ($U_L$) as a function of $G_{ad}(OOH^*)$ was displayed in the activity volcano plot (Fig. 5f). Most configuration models except CoN$_2$O$_2$-C were located at the left side of the volcano plot (Fig. S24), demonstrating the stronger adsorption of OOH* and the reduction of OOH* to $H_2O_2$ was the rate-determining step[36]. Moreover, due to the linear scaling relationship between $G_{ad}(OOH^*)$ and $G_{ad}(O^*)$, the stronger $G_{ad}(OOH^*)$ corresponds to the weaker dissociation of $H_2O_2$ ($G_{ad}(H_2O_2) \cdot G_{ad}(O^*)$), thus exhibiting higher $H_2O_2$ selectivity (Fig. S25a). In addition, the geometric asymmetry of different configurations was further evaluated by calculating the distance between the Co atom and the graphene plane (defined as the off-center distance), which was further used as a function to describe the $G_{ad}(OOH^*)$ in Fig. 5g. A positive correlation was observed between the off-center distance of the Co atom and $G_{ad}(OOH^*)$ of asymmetric configurations. Furthermore, the projected density of states (PDOS) of Co 3d orbitals of the examined configuration models were calculated to further investigate the electronic properties and reveal the underlying relationships (Fig. S25b). Compared with CoN$_4$ configuration, all the asymmetric configurations exhibited decreased occupation states near Fermi level and more negative d-band centers, indicating a weaker adsorption strength for intermediates (Fig. S25c). Moreover, the positive relationship between negative d-band shift center and off-center distance was found, demonstrating that more distorted configuration presented weaker adsorption of intermediates (Fig. S25d). Nevertheless, apart from the d-band center theory, other factors such as additional carbon layer (Fig. S26), might also contribute to adjusting the adsorption energy of intermediates, which appealed to more consideration in the future discussion.

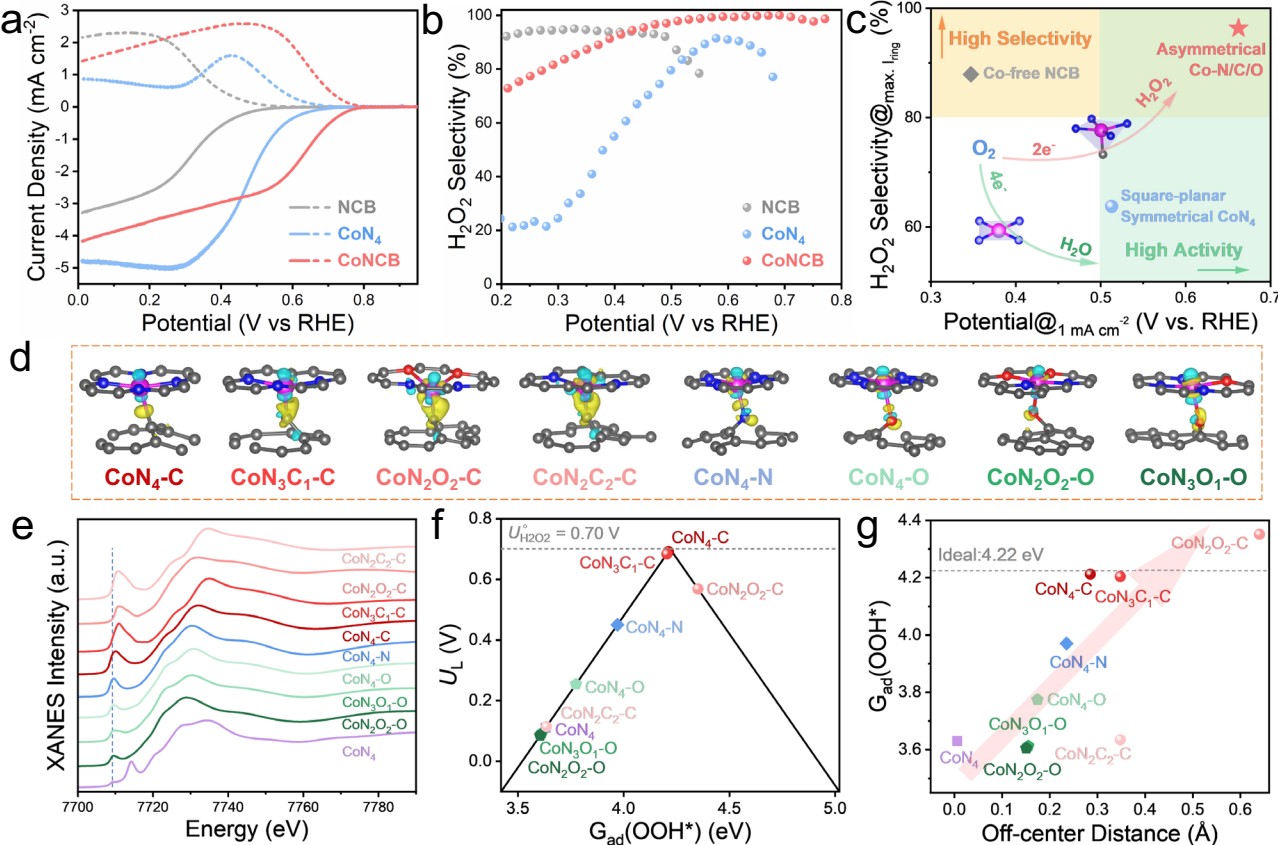

**Fig. 5 | DFT calculations. a** ORR polarization curves of disk current density (solid line) and ring current density (dash line) of NCB, $CoN_4$, and CoNCB in 0.1 M PBS (pH = 7). **b** Calculated $H_2O_2$ selectivity ($H_2O_2$ %). **c** Comparison of $H_2O_2$ selectivity at the potential of the maximum ring current density. **d** Differential charge density of various Co-C/N/O models. The isosurface level is 0.01 e/bohr³. The yellow and blue region denotes accumulation and decrease of electron density respectively. More views of examined configurations of possible Co-C/N/O species are provided in Fig. S22. **e** Related simulated XANES spectra. **f** Calculated ORR activity volcano plot. The dashed black line represents the equilibrium potential of the theoretical 2e⁻ ORR. **g** Plot of $G_{ad}(OOH^*)$ as a function of the off-center distance of Co atom. The dashed black line represents the ideal $G_{ad}(OOH^*)$ of 4.22 eV.

## Discussion

In summary, we first revealed the significance of EHPP study in neutral media and highlighted the challenges associated with designing and charactering of traditionally annealed M-N-C catalysts. Based on this background, the surface engineering strategy was applied to the synthesis of a series of M-N-C electrocatalysts with ultra-low loading of 3d transition metals. The atomic dispersion of M-C/N/O species was evidenced by STEM and XAFS analysis in both local and bulk perspectives. Moreover, the comprehensive analysis including Co L-edge NEXAFS, Co K-edge XANES, and FT-EXAFS demonstrated the asymmetric electronic configuration of the Co-C/N/O species in the CoNCB material. Electrochemical evaluation of the CoNCB material showed excellent EHPP performance, i.e., the positive onset potential of 0.76 V (at 0.1 mA cm⁻²) and high $H_2O_2$ selectivity of over 95 % (0.45−0.75 V) were achieved. Furthermore, ultrahigh mass activity ($6.1 \times 10^5$ A $g_{Co}^{-1}$ at 0.5 V) was obtained. The practical application of CoNCB material was evaluated using a flow cell with the $H_2O_2$ production rate reaching 4.72 mol $g_{catalyst}^{-1}$ h⁻¹ cm⁻². To further understand the correlation between Co-C/N/O configurations and observed performance, experimental and computational studies were conducted, highlighting the superiority of asymmetric Co-C/N/O configurations over the square-planar symmetric Co-N₄ configuration. Notably, our findings revealed a positive correlation between the enhanced of $G_{ad}(OOH^*)$ and the off-center distance of the Co atom. We believe that our work provides novel perspectives and methodological guidelines for evaluating the electronic configurations of catalysts and advancing the field of EHPP in neutral media.

## Method

### Chemicals and materials

Manganese(II) nitrate tetrahydrate ($Mn(NO_3)_2 \cdot 4H_2O$), manganese(II) phthalocyanine (MnPc), iron(III) nitrate nonahydrate ($Fe(NO_3)_3 \cdot 9H_2O$), iron(II) phthalocyanine (FePc), cobalt(II) nitrate hexahydrate ($Co(NO_3)_2 \cdot 6H_2O$), 2,3,7,8,12,13,17,18-Octaethyl-21H,23H-porphine cobalt(II) (CoPr), nickel(II) nitrate hexahydrate ($Ni(NO_3)_2 \cdot 6H_2O$), nickel(II) phthalocyanine (NiPc), copper(II) nitrate trihydrate ($Cu(NO_3)_2 \cdot 6H_2O$), copper(II) phthalocyanine (CuPc), potassium phosphate monobasic ($KH_2PO_4$), potassium phosphate dibasic ($K_2HPO_4$), perchloric acid ($HClO_4$), potassium hydroxide (KOH), nafion perfluorinated resin solution (5 wt%), hydrochloric acid (HCl) were purchased from Sigma-Aldrich (UK) Co., Ltd. Ketjenblack EC300J carbon black (CB) was purchased from AkzoNobel Co., Ltd.

### Materials synthesis

Synthesis of OCB. To prepare the oxidized carbon black (OCB), 1.5 g CB was dispersed in 100 mL 35 wt% $HNO_3$ solution and heated at 80 °C under reflux with stirring for 15 h. After cooling to room temperature, the suspension was washed thoroughly to pH neutral by centrifuging using excess ultrapure water. The collected precipitate was dried for 48 h in the freeze dryer (Virtis Benchtop Pro) to obtain the OCB.

Synthesis of CoNCB. To prepare cobalt-nitrogen doped carbon black (CoNCB), 100 mg OCB was dispersed into 30 mL deionized water and ultra-sonicated (Fisherbrand™ 505 Sonicator) for 30 min in the ice bath. Subsequently, 370 μL of 1 M $Co(NO_3)_2 \cdot 6H_2O$ was dissolved into the suspension and further ultra-sonicated in the ice bath for 30 min.

After that, the suspension was frozen by liquid nitrogen immediately and freeze-dried for 72 h. Then, the as-obtained powder was annealed in NH₃ at 850 °C for 1 h in the tube furnace with a heating rate of 10 °C/min. Afterwards, the collected carbon powder was stirred in 4 M HCl overnight. Finally, the carbon powder was washed with excess ultra-pure water by centrifuging and freeze-dried for electrochemical characterization.

MnNCB, FeNCB, NiNCB, and CuNCB were prepared in the same way except using different metal precursors, which are 320 μL of 1 M Mn(NO₃)₂·4H₂O, 510 μL of 1 M Fe(NO₃)·9H₂O, 370 μL of 1 M Ni(NO₃)₂·6H₂O, 300 μL of 1 M Cu(NO₃)₂·6H₂O, respectively. Nitrogen-doped carbon black (NCB) was also prepared in the same way without using any metal precursor.

## Materials characterization

TEM samples were prepared by drop casting a sample/ethanol suspension onto copper TEM grids with lacey carbon film. The grids were then dried under vacuum at 80 °C for 15 min. The morphology of the samples was investigated on the aberration-corrected scanning transmission electron microscope (JEOL ARM300CF) at the electron Physical Science Imaging Centre (ePSIC) E02 beamline of Diamond Light Source (UK). A 46.5 pA probe with a 31 mrad convergence semi-angle was used at 80 keV acceleration voltage. ABF (annular bright field) and HAADF (high-angle annular dark field) signals were acquired with collection semi-angles of 12−26 mrad and 74−155 mrad, respectively.

The metallic content of Mn, Fe, Co, Ni, and Cu was determined using microwave plasma atomic emission spectroscopy (MP-AES, Agilent 4210 MP-AES). The carbon materials were first burned in the air at 850 °C for 3 h with a heating rate of 3 °C/min in a quartz tube, respectively. The collected ash was dissolved into 10 mL 4 wt% HNO₃ solution by sonicating. Standard samples with concentrations of 0, 4, 8, 12, 16, and 20 ppm were prepared first to perform the calibration.

The near-edge X-ray absorption fine structure (NEXAFS) measurements in the soft X-ray regime were performed at Branch B of the VerSoX beamline B07 of Diamond Light Source (UK)[30,50]. The storage ring was operated at 3 GeV with a ring current of 300 mA[51]. The beam size at the sample was approximately $0.1 \times 0.1$ mm² (V x H) and the photon flux was ~10¹⁰ ph/s. Data was collected at O K-edge (510−585 eV), N K-edge (385 − 440 eV), and Co L-edge (755 − 815 eV) in total electron yield (TEY) mode under vacuum ($<1 \times 10^{-7}$ mbar). Samples were prepared by drop-casting onto gold-coated silicon wafers using deionized water as the solvent. At least 5 repetitions of NEXAFS scans were collected and averaged to improve the signal-to-noise ratio for each sample. The photon energy was calibrated based on the known features of incident beam flux (I₀) and/or NEXAFS spectra of standard reference samples. The spectra were divided by I₀; background was subtracted at the low photon energy side, below the energy of the first absorption feature; all spectra were normalized to 1 at a high photon energy above the highest resonance.

Mn, Fe, Co, Ni, and Cu K-edges X-ray absorption spectroscopy (XAS) data were collected at B18 beamline of Diamond Light Source (UK)[52]. A monochromatic beam between 6.34 keV and 9.98 keV was producing using Si(111) double crystal monochromator and the rejection of higher harmonics was achieved by using two dedicated Pt coated mirrors operating at 7 mrad incidence angle. The beam size at the sample was approximately $1.0 \times 1.0$ mm² (V x H) and the photon flux was ~10¹¹ ph/s (no attenuation). The XAS spectra were collected in transmission mode and fluorescence mode. Intensity of the incident beam (I₀) and the transmitted beam (Iₜ) were monitored by ionization chambers (filled with a mixture of He, N₂, and Ar gases). Fluorescence measurements were performed using Canberra 36-pixel Monolithic Segmented Hyer Pure Germanium Dector (HPGe) partnered with the Xspress4 digital pulse processor[53]. For fluorescence

measurements, M-N-C powdered samples were pressed into pellets (diameter: 8 mm), while for transmission measurements, standard reference samples were diluted with cellulose before pressing into pellets (diameter: 8 mm).

The XAS spectra of each sample were measured at least 10 times for fluorescence mode and 3 times for transmission mode under room temperature and merged to improve the signal-to-noise ratio. Metal foil was measured simultaneously for each sample as a reference for energy calibration. XAS data were analyzed using the Demeter software package (including Athena and Artemis, version 0.9.26)[54]. The k²-weighted wavelet transform EXAFS (WT-EXAFS) was generated using Hama software (Morlet function)[55].

## Electrochemical tests

All the electrochemical tests were carried out in a three-electrode cell system controlled by a Gamry electrochemistry station. The rotating ring disk electrode (RRDE) working electrode (Pine E7R9) comprises a glassy carbon rotation disk electrode (0.2475 cm² area) and a Pt ring (0.1866 cm² area) with a theoretical collection efficiency (N) of 37%. An Ag/AgCl electrode and a graphite rod were used as the reference electrode and counter electrode, respectively. 0.05 μm Al₂O₃ polishing suspension was used to clean the working electrode. To prepare the ink, 1 mg of catalyst was dispersed in 1.94 mL of ethanol and 60 μL of 5.0 wt% Nafion solution. After ultrasonication in the ice bath for 30 min, 2 μL of suspension was drop-casted onto the disk electrode on the rotating stage and dried at room temperature.

The oxygen reduction activity and selectivity of the samples were evaluated by RRDE measurements in O₂-saturated 0.1 M phosphate buffered solution (PBS, pH = 7). Cyclic voltammetry (CV) was first performed at a scan rate of 100 mV s⁻¹ between 0.164 and 0.964 V vs. RHE in O₂-saturated electrolytes until a steady CV was obtained. After that, RRDE measurement was carried out in the O₂-saturated electrolyte between 0.164 and 0.964 V at a scan rate of 10 mV s⁻¹ under 1600 rpm. The potential of the Pt ring electrode was set to 1.2 V vs. RHE. To eliminate the formation of PtOₓ, chronoamperometry was carried out at 0 V for 30 s prior to the test.

All electrode potentials were calibrated to RHE without IR correction using Eq. (1):

$$E_{RHE} = E_{Ag/AgCl} + 0.197 + 0.059\,pH \tag{1}$$

The H₂O₂ selectivity was determined based on Eq. (2):

$$H_2O_2\,\% = 200 \times \frac{I_{Ring}/N}{I_{Disk} + I_{Ring}/N} \tag{2}$$

where $I_{Ring}$ is the ring current, $I_{Disk}$ is the disk current and $N$ is the collection efficiency. The experimental $N$ was determined according to our previous work and the experimental value is 0.375[36].

Tafel slopes were evaluated according to the Koutecky-Levich equation (Eq. 3):

$$\frac{1}{J_{H_2O_2}} = \frac{1}{J_{K,H_2O_2}} + \frac{1}{J_{L,H_2O_2}} \tag{3}$$

where $J_{H_2O_2}$ is the measured ring current density. $J_{L,H_2O_2}$ is the theoretical limiting current of the 2e⁻ ORR process, which is determined to be 2.9 mA cm⁻². $J_{K,H_2O_2}$ is the kinetic current of H₂O₂[56].

Turnover frequency (TOF) was determined based on Eq. 4:

$$TOF = \frac{M_{Co}J_{H_2O_2}}{n \cdot F \cdot m \cdot C_{Co}} \tag{4}$$

where $M_{Co}$ is the atomic weight for Co (58.93 g/mol), $n$ is the number of electrons transferred (the value of $n$ is 2 for 2e⁻ ORR), $F$ is the Faraday

constant ($F = 96485$ C/mol), $m$ is mass loading, $C_{Co}$ is the content of Co in the catalysts determined by MP-AES.

## Bulk electrolysis

Bulk electrochemical $H_2O_2$ production was carried out in a two-compartment flow cell separated by NR-212 membrane (Sigma-Aldrich). The CoNCB material was deposited on the gas diffusion electrode (Sigracet 29 BC) using the air-brushing method. The actual area was 0.25 cm$^2$ with a loading of 0.2 mg/cm$^2$. The nickel foam was used as the anode and Ag/AgCl electrode was used as the reference electrode. 1 M PBS (pH = 7) solution was used as the recycled electrolyte for both anode and cathode compartments. The flow rate of the cathode compartment was 50 mL min$^{-1}$ while it was 100 mL min$^{-1}$ for the anode compartment. The $O_2$ supply rate was around 50 mL min$^{-1}$ to the cathode. The electrochemical $H_2O_2$ production was performed at a current of 20 mA cm$^{-2}$.

The cerium sulfate $Ce(SO_4)_2$ titration method was used to measure the produced $H_2O_2$ concentration. The concentration of $H_2O_2$ can be calculated by Eqs. (5–6). The Faradaic efficiency for $H_2O_2$ production was calculated by Eq. (9). Different concentrations of $Ce(SO_4)_2$ (up to 0.5 mM) were prepared in 0.5 M $H_2SO_4$ solution. The absorbance intensity at 319 nm in UV spectroscopy of known concentration (0, 0.1, 0.2, 0.3, 0.4, 0.5 mM) of $Ce^{4+}$ solution was linearly calibrated, as shown in Fig. S20.

For bulk electrolysis in 1 M PBS, 100 μL of solution in the recycled electrolyte of the cathode was sampled and injected immediately into 9.9 mL of 0.5 mM $Ce^{4+}$ solution periodically.

$$2Ce^{4+} + H_2O_2 \rightarrow 2Ce^{3+} + 2H^+ + O_2 \qquad (5)$$

$$M(H_2O_2) = 1/2\,M(Ce^{4+}) \qquad (6)$$

where $M(Ce^{4+})$ is the mole of consumed $Ce^{4+}$.

$$\text{Faradaic efficiency (\%)} = \frac{2 \times 96485 \times M(H_2O_2)}{\int_0^t I\,dt} \times 100\% \qquad (7)$$

where $\int_0^t I\,dt$ is the cumulative charge.

## Computational methods

The Vienna Ab Initio Simulation Package (VASP) was used to perform periodic spin-polarized periodic density functional theory calculations[57–59]. The projector augmented wave (PAW) method was employed and for the expansion of the plane-wave basis sets, the cut-off energy was set to 400 eV. The Perdew-Burke-Ernzerhof (PBE) version of the exchange and correlation function was used to relax the structures and perform the total energy calculations[60]. In this study, we also included that Grimme's dispersion correction (DFT + D3) as dispersive effects may play a significant role in these systems under investigation[61]. For all the calculations, in the direction perpendicular to the surface, we used a vacuum gap of ~15 Å, which was sufficient to eliminate any spurious interactions along the z-axis.

The adsorption free energy ($G_{ad}$) of the OOH* species on the GNR models was calculated using:

$$G_{ad}(OOH^*) = E_{\text{model}+OOH^*} - E_{\text{pristine model}} - 2E_{H_2O} + \frac{3}{2}E_{H_2} + \Delta E_{ZPE} - T\Delta S \qquad (8)$$

where, $E_{\text{model}+OOH^*}$, $E_{\text{pristine model}}$, $E_{H_2O}$, $E_{H_2}$, $\Delta E_{ZPE}$, and $\Delta S$ are the total energies of the OOH* species adsorbed on GNR, pristine GNR model, the $H_2O$ molecule, the $H_2$ molecule, zero-point energy change and entropy change, respectively.

For 2e$^-$ ORR, there were generally two coupled electron and proton transfers:

$$O_2 + {}^* + (H^+ + e^-) \rightarrow OOH^* \qquad (9)$$

$$OOH^* + (H^+ + e^-) \rightarrow H_2O_2 + {}^* \qquad (11)$$

The computational hydrogen electrode (CHE) model was employed to evaluate the Gibbs reaction-free energy change (ΔG) for each step in the 2e$^-$ ORR[49]. The free energy for each reaction intermediate was defined by Eq. (12):

$$\Delta G = \Delta E_{DFT} + \Delta E_{ZPE} - T\Delta S + eU \qquad (12)$$

$\Delta E_{DFT}$ is the adsorption energy calculated by DFT. $\Delta E_{ZPE}$ is the zero-point energy change and ΔS is the entropy change at 300 K. T, e, and U are the temperature, the number of electrons transferred, and the electrode potential, respectively.

The crystal orbital Hamilton population (COHP) analysis was performed using Local Orbital Basis Suite Toward Electronic Structure Reconstruction (LOBSTER) package[62].

FDMNES package was used to simulate the Co K-edge XANES for proposed models in the framework of a real-space full multiple-scattering scheme with the muffin-tin approximation[63,64]. The energy-dependent exchange–correlation potential was calculated in the real Hedin–Lundqvist scheme, and then the XANES spectra were convoluted using a Lorentzian function with an energy-dependent width to account for the broadening from both the core-hole and final-state widths. A cluster of 5.0 Å radius containing ~48 atoms was used in the calculation with self-consistency.

## Data availability

The authors declare that all data supporting the findings of this study are provided within the article and Supplementary Information. All data will be available from the corresponding author upon request.

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

## Acknowledgements

The project is funded by EPSRC (EP/L015862/1; EP/V027433/3) and UK Research and Innovation (UKRI) under the UK government's Horizon Europe funding guarantee (101077226; EP/Y008707/1) for funding support. We acknowledge electron Physical Science Imaging Centre instrument E02 (session ID: MG30614, MG32058, MG32035), B18 beamline (session ID: SP30595), B07-B beamline (session ID: SI32117, SI33466) at Diamond Light Source for the allocated experiment session and the UK Catalysis Hub block allocation for beamtime (SP29271). L.L. would like to thank the funding support from China Scholarship Council/University College London for the joint Ph.D. scholarship.

## Author contributions

L.L. and G.H. conceived the project and designed the experiments; L.L. synthesized and characterized the materials; L.L., L.K., Y.T., H.G., I.M., V.C., D.G., J.C., and M.W. performed XAFS measurement and analysis; J.F. performed DFT calculations; L.L., D.G.H., and C.A. performed STEM measurements; L.L., L.K., T.W., L.Z., K.L., G.H., P.F., and D.G. performed NEXAFS measurements; L.L., J.Zhang., and J.Zhu. performed the flow cell measurements; L.L., J.Z., and S.C. performed the MP-AES measurement; I.P. and G.H. supported this project; The manuscript was written through contributions of all authors.

## Competing interests

The authors declare no competing interests.
