## [Peer Review File · Nature Communications]

Atomically Dispersed Asymmetric Cobalt Electrocatalyst for Efficient Hydrogen Peroxide Production in Neutral MediaREVIEWER COMMENTS

Reviewer #1 (Remarks to the Author):

The contribution by Liu at all discusses a new cobalt based electrocatalyst and its performance in hydrogen peroxide synthesis. The paper includes characterization and performance sections. I'd like to address them separately

Characterization:

1. However hard one tries, it's hard to discern single atoms in figure 1b. One can see some isolated spot but there are also bright areas across the image. Are these area cobalt atoms aggregation? Atomic dispersion cannot be confirmed with the quality of the image. This is not to say Co is not single-atom dispersed – just that the image is poor evidence of it. See e.g., Fig 1 in 10.1039/D1EE00878A for a TEM image that can support a atomic dispersion claim.
2. Soft X-ray data is presented but either the quality or complexity of the spectra (Co and O respectively) does not allow to draw any conclusions. Authors indicate that "spectra demonstrated the complicated presence of O and N species" – but it doesn't warrant including the data in the manuscript.
3. CoNCB is certainly highly heterogeneous material and the conclusions that can be drawn from XANES and EXAFS analysis are limited, as those are averaging techniques. The XANES spectrum in Figure 2d can easily result from a combination of Co oxides. Indeed a strong feature of the planar Co in CoPr is absent in CoNCB, but the only insight from this data is that Co in CoNCB is not planar.
4. Wavelet analysis is hardly conclusive either. Authors make a statement that "WT EXAFS provides the straightforward way to discriminate the backscattering atoms in the same shell" which is a popular but disputed claim. It is not clear how they arrive to Co-C/N/O from the position of the peak in the wavelet transform map. Again, the only insight here is that Co environment in CoNCB is different from CoPr.
5. EXAFS analysis did not yield any insights, most likely due to heterogeneity of Co sites in the materials. I praise the authors for being straightforward on it. The literature in this field is littered by structures derived from EXAFS without considering alternatives. Yet to add to my point – due to the complexity and heterogeneity of the material characterization does not deliver any information that cannot be inferred from literature on the topic.

With rather inconclusive characterization section, the manuscript claim to fame is the excellent performance of the catalysts in hydrogen peroxide synthesis. With so many recent studies reporting "record" performance I with the community could agree on the universal metrics and units before making "highest activity and selectivity" claims. For example, a publication by Cao et al earlier this year (10.1038/s41467-023-35839-z) claims "record rates" in similar materials but the units in two papers are different – and Cao's paper is not cited in the current manuscript. This is a systemic problem, but it makes it difficult to evaluate the paper.

With these deficiencies the publication of the manuscript in Nature Communication is not warranted. My suggestion is to rework the paper and submit it to a more specialized journal.

Some minor remarks:

Line 162 – perhaps "broadened peaks"

Line 195 – "red shift" is optical spectroscopy jargon. Shift to the lower energy is perhaps more descriptive

Line 202: FT EXAFS is not a "process"

Line 253: replace "inclined" by "increased"

Line 374: I simply do not understand this sentence

Reviewer #2 (Remarks to the Author):

The authors reported a series of M-N-C electrocatalysts at ultralow metal-loading for O₂-to-H₂O₂ conversion within neutral electrolyte. Specifically, CoNCB with the asymmetric Co-C/N/O coordinative configuration exhibited the highest electrocatalytic activity and selectivity. To shed light on the structure-performance relationship, comprehensive X-ray radiated characterizations have been carried out, pinpointing the key role of asymmetric Co-C/N/O configuration in promoting H₂O₂ electrosynthesis. To this reviewer, the manuscript is well organized, the electronic structure of prepared catalysts is well characterized by the state-of-the-art synchrotron techniques. I'd suggest its acceptance at Nat. Commun. after addressing the below concerns:

1. The authors should double-check the assignment of pyrrolic N peak in N K-edge NEXAFS, as the observed peak position is in controversial to the cited Refs. 36&37.
2. Regarding to the clarification of M-X-C coordination environment within practical single atom catalysts as well as its impact on branching electrocatalytic pathways, I'd direct the authors to some closely related works at 10.31635/ccschem.020.202000667 and 10.1038/s41467-022-28346-0. Proper remarks should be supplemented.
3. In RRDE measurements, the catalyst loading is ca. 4 $\mu\text{g}/\text{cm}^2$. Proper justification should be provided for this ultra-low catalyst usage, actually it's even hard to fully cover the glassy carbon disk at this dosage.
4. Fig. 3d plots the experimentally determined Tafel slopes for various M-N-C catalysts, where a deepened discussion on the relevant rate-determining step is encouraged, in conjunction with the theoretical simulation part.

Reviewer #4 (Remarks to the Author):

The authors reported on surface-engineered M-N-C, which exhibited high selectivity and activity for the 2e⁻ ORR in a neutral medium. Various characterization methods and DFT simulations were employed to investigate the experimental observations. I found this manuscript to be well-written and interesting. I recommend a major revision.

1. The author stated that "The Gibbs free energy diagram showed the trend of the kinetic barrier at the equilibrium potential of 0.7 V for various models." However, in Figure 5f, it depicts only the energy of the states at 0.7 V, without showing the kinetic barrier. The protonation calculations using explicit water molecules and constant potential calculation scheme are more than 100 times computationally expensive than mere thermodynamic calculations.
2. While Figures 5a and 5b shows H₂O₂ activity and selectivity of CoNCB, respectively, DFT calculation results exclusively highlight the activity for the 2e⁻ ORR. Additional analysis on H₂O₂ selectivity is required.
3. The author's explanation for the positive correlation between the off center distance and Gad(OOH*) is insufficient. At least, an electronic structure analysis is required to provide the underlying reasons.
4. Despite of highlighting the reaction in a neutral media, the DFT calculations were performed under the vacuum system. Recently, the constant potential calculation using implicit solvation method is becoming a standard in the field. Since this method can include the effect of pH, cation, etc, it is

highly recommended to perform this method.

5. The configurations of CoNCB were modelled with an additional carbon layer for axial ligand. Is the enhanced performance result from an axial ligand or an additional carbon layer or both? The merit of DFT calculations is that several effects can be separated by properly modelling the simulations.

Reviewer 1

The contribution by Liu at all discusses a new cobalt based electrocatalyst and its performance in hydrogen peroxide synthesis. The paper includes characterization and performance sections. I'd like to address them separately.

Response: We appreciate the reviewer's insightful comments, especially regarding X-ray adsorption spectroscopy characterization. Point-by-point responses and explanations are provided accordingly. We hope these could address your concerns and look forward to any further feedback.

Characterization:

1. However hard one tries, it's hard to discern single atoms in figure 1b. One can see some isolated spot but there are also bright areas across the image. Are these area cobalt atoms aggregation? Atomic dispersion cannot be confirmed with the quality of the image. This is not to say Co is not single-atom dispersed – just that the image is poor evidence of it. See e.g., Fig 1 in 10.1039/D1EE00878A for a TEM image that can support a atomic dispersion claim.

Response: We appreciate the reviewer pointing this out. The astigmatism of HAADF-STEM image of CoNCB in Figure 1d (in the first version) was not precisely corrected. To address this, we re-measured this sample with good beam alignment and precise astigmatism correction. STEM images of CoNCB and NCB with both HAADF and ABF modes are shown in **Figure R1.1a-b**. The HAADF-STEM image of CoNCB displays distinct contrast between cobalt single atoms and carbon substrate. These images have also been updated in the revised manuscript.

Concerns regarding the bright areas and potential cobalt atoms aggregation can be addressed based on the principles of HAADF-STEM images (*J. Anal. Sci. Technol*, 2018, 9, 1-14; *Physical principles of electron microscopy*. 2005, New York: Springer.). The bright spots are formed from signals of scattering electrons collected by the high-angle annular dark field detector. Atoms with larger Z number and heavier atomic mass present higher scattering electron angle, thus distinguishing Co atoms and carbon substrate in CoNCB material. Additionally, the signals are dependent on the density and thickness of specimen. Dense carbon lattice and thick area possess higher scattering angle than thin amorphous carbon substrate. This is exemplified by the HAADF-STEM image of NCB without metal doping in **Figure R1.1d**, where carbon lattice is shown in bright. Additionally, the HAADF-STEM images reflect the 2D projections of the 3D objects. Thus, Co atoms locate at same area but with different heights would exhibit the 'aggregation' features. Therefore, it is easier

to discern Co atoms in amorphous and flat areas than dense spherical areas.

We appreciate that reviewer providing a good reference (*Energy Environ. Sci.*, 2021, 14, 10, 5444-5456 (10.1039/D1EE00878A)), which shows a high-quality of HAADF-STEM image (as shown in **Figure R1.2**). We assumed that there are two reasons contributing to the good quality of the HAADF-STEM image in this reference. Firstly, Co loading in their Co-N-C sample (0.97 wt%) is much higher than that of our CoNCB sample (0.05 wt%), making it easier to observe massive Co atoms. Secondly, according to the synthesis method in this reference, Co-N-C SAC was prepared by annealing the triquinoxalinylene–Co coordination and carbon black complex, thereby forming abundant amorphous graphite layers wrapping on the carbon black support (**Figure R1.2b**). In contrast to the dense carbon layer on the carbon black, it was easier to discern Co atoms on the amorphous graphite layer using HAADF-STEM. In addition, the white area in **Figure R1.2c** is ascribed to the graphite layer wrinkle, and the aggregation Co atoms around the wrinkle is also difficult to judge due to the interference of dense carbon layer and the 2D projections issues.

Therefore, we hope that our response could address the reviewer's concerns regarding the STEM characterization of atomically dispersed Co species.

Figure R1.1. (a) ABF-STEM image and (b) HAADF-STEM image of CoNCB. (c) ABF-STEM image and (d) HAADF-STEM image of NCB.

Figure R1.2. (a) Schematic synthesis of a nitrogen coordinated cobalt single-atom catalyst (Co–N–C), (b) high-angle annular dark-field scanning transmission electron microscopy (HAADF-STEM) image with corresponding energy dispersive spectroscopy (EDS) elemental mappings of C, N and Co, (c) aberration-corrected HAADF-STEM image of Co–N–C, (d) the enlarged colored image with isolated Co single atoms marked with blue. Copyright 2021, The Royal Society of Chemistry (*Energy Environ. Sci.*, 2021, 14, 10, 5444-5456).

2. Soft X-ray data is presented but either the quality or complexity of the spectra (Co and O respectively) does not allow to draw any conclusions. Authors indicate that “spectra demonstrated the complicated presence of O and N species” – but it doesn’t warrant including the data in the manuscript.

Response: We agree with the reviewer that the signal-to-noise ratio of Co L-edge spectrum is poor to draw specific conclusions, and K-edge spectra of O and N species are very complicated. Nevertheless, we would like to explain why conduct the soft X-ray measurement. Moreover, additional information about N K-edge measurement has been provided.

1. Soft X-ray data of Co L-edge spectra corresponding to the dipole-allowed 2p-3d electron transition provide supplementary information of oxidation state and electronic structures to the hard X-ray data of Co K-edge. In contrast to roughly determining the oxidation state from the energy shift of XANES rising edge in previous research (*Nat. Commun.*, 2021, 12, 5589; *Nat. Commun.*, 2021, 12, 6806; *Nat. Commun.* 2023, 14, 6666; *Nat. Commun.*, 2023, 14, 3776), the combination of K-edge and L-edge analysis provided a more profound evaluation method to determine the oxidation and electronic state of transition metals in

M-N-C materials. We intended to offer a good example. Unfortunately, the Co L-edge signal in CoNCB material is very weak to determine the electronic structure due to the ultra-low Co loading (although tens of scan repetitions were merged). Nevertheless, we still observed that Co L-edge spectrum of CoNCB displays different features compared to cobalt porphyrin reference with specific square-planar CoN₄ configuration, indicating the square-planar CoN₄ configuration is absent in CoNCB material. In addition, similar to the hard X-ray data analysis, the complexity in identifying M-N-C configurations results from the heterogeneity formed in high-temperature annealing process.

2. O K-edge spectra were provided to illustrate the presence of oxygen functional groups on the surface of CoNCB material. Different with popularly proposed Co-N_x (x = 2-5) configurations (*J. Am. Chem. Soc.* 2020, 142, 39, 16861–16867; *J. Am. Chem. Soc.* 2022, 144, 32, 14505–14516; *Energy. Environ. Sci.*, 2021, 14, 10), 5444-5456), O K-edge characterization offers the possibility of Co-O coordination, which introduces the complexity in subsequent FT-EXAFS fitting to determine the accurate coordination environment.
3. For the same reason, we provided the N K-edge spectra here. Notably, we just found that the N K-edge spectrum was contaminated in our recent beamtime in October 2023. Obvious N K-edge features were found on the clean bare gold substrate (**Figure R1.3**). The features at around 398.3 eV, 400.2 eV, 402.8 eV, 406.7 eV and 412.4 eV partly arise from the contamination in the beamline set-up. Silicon window was used to measure this sample at B07-B beamline at Diamond Light Source to avoid the N K-edge interference from silicon nitride window (B07-B beamline setup). The contamination was probably due to the glue used for the silicon window assembly after discussing with beamline scientists. We sincerely apologize for this as we cannot find a good way to obtain the real N K-edge at the moment. We have added additional supporting note to the revised supporting information to prevent causing misleading understanding.

Figure R1.3. N K-edge NEXAFS spectra of CoNCB and bare gold substrate. The N K-edge feature on bare gold substrate was due to the contamination in the beamline setup.

3. CoNCB is certainly highly heterogeneous material and the conclusions that can be drawn from XANES and EXAFS analysis are limited, as those are averaging techniques. The XANES spectrum in Figure 2d can easily result from a combination of Co oxides. Indeed a strong feature of the planar Co in CoPr is absent in CoNCB, but the only insight from this data is that Co in CoNCB is not planar.

Response: We agree with the reviewer that XANES and EXAFS are averaging techniques. We underscored this in our manuscript that they are collected ‘based on the average bulk information’. We also pointed out that the spectrum might result from a combination of Co-C/N/O species. However, we disagree with that the XANES spectrum results from a combination of Co oxides in CoNCB material. Here are two reasons: 1. Concerning synthesis route, Co loading is very low (0.05 wt%), and the anchoring of Co species mainly occurred on the carbon black surface during the reaction. Afterwards, acid washing was conducted to remove any possible residual cobalt oxides, thus remedying the possibility of Co oxides forming in the CoNCB material. 2. In terms of the XAFS characterization, if Co oxides exist, the oscillation of Co-Co scattering in the second shell (2-3 Å in R space without phase correction) is very strong due to the ordered structure of Co oxides (As shown in **Figure R1.4b**). However, this was not observed in the FT-EXAFS spectra of CoNCB. FT-EXAFS spectra of synthesized CoOx clusters can be referred to the following publications: *Energy Stor. Mater.*, 2020, 29, 156-162; *Adv. Energy Mater.*, 2022, 12, 26, 2200716. In addition, no nanoparticles of Co oxides were observed in the TEM image, even though TEM could only provide the local area information.

Figure R1.4. (a) Co K-edge XANES spectra. (b) FT magnitudes of k^2 -weighted EXAFS spectra without phase correction. Spectra are displaced vertically for clarity.

Regarding the insight of XANES analysis, we appreciate the reviewer agreeing with us that the configuration of Co specie in CoNCB is not planar. Unlike previous reports that square planar Co-N₄ or other Metal-N₄ configurations were proposed only supported by FT-EXAFS fitting (*J. Am. Chem. Soc.* 2022, 144, 32, 14505–14516; *Appl. Catal. B: Environ.*, 2023, 324: 122267; *Energy. Environ. Sci.*, 2022, 15(6): 2619-2628; *Nat. Catal.* 2021, 4, 615–622), we drew attention to XANES analysis that the planar configurations were not present in fact. In addition, we also pointed out that the determination of oxidation state resulting from evaluating the energy shift of XANES was not feasible due to the complicated orbital hybridization in the system of Co-C/N/O species (*ACS Catal.* 2022, 12, 10, 5864–5886). Since the configuration of Co species in CoNCB is not planar, we carried out the DFT study to further investigate the relationship between multiple possible configurations and the reaction behaviors.

4. Wavelet analysis is hardly conclusive either. Authors make a statement that “WT EXAFS provides the straightforward way to discriminate the backscattering atoms in the same shell” which is a popular but disputed claim. It is not clear how they arrive to Co-C/N/O from the position of the peak in the wavelet transform map. Again, the only insight here is that Co environment in CoNCB is different from CoPr.

Response: We thank the reviewer pointing this out. We deleted this disputed claim in the revised manuscript. Although WT-EXAFS could discern the contribution of each pathway in R-space and k-space at the same time, it is not feasible to separate the close and complex Co-C/N/O coordination environments. Besides, considering WT-EXAFS results from k-space of Co K-edge and the possible coordination atoms in the first shell are within C, N, or O atoms, we ascribed the peak in the WT-

EXAFS contour map to the contribution of complex Co-C/N/O species but not individual configuration. Related description was revised as following:

‘In addition, WT-EXAFS spectra are provided to provide the R-space and k-space at the same time. In the R-k contour map (Figure 2g-d), Co foil reference showed the maximum intensity at $k = 7.1 \text{ \AA}^{-1}$ and $R = 2.2 \text{ \AA}$, corresponding to Co-Co contribution. Due to the low atomic number of C/N/O, CoPr reference material exhibited the maximum intensity at $k = 4.6 \text{ \AA}^{-1}$ and $R = 1.5 \text{ \AA}$ corresponding to the Co-N contribution. As for the CoNCB catalyst, the maximum intensity appeared at $k = 3.1 \text{ \AA}^{-1}$ and $R = 1.4 \text{ \AA}$, which is far away of the intensity of Co-Co scattering at $k = 7.1 \text{ \AA}^{-1}$, indicating the Co-C/N/O contributions. Of note, WT-EXAFS spectrum is not feasible to discern the contribution of individual Co-C, Co-N, or Co-O coordination due to the approximate k-space dependencies.^{46, 47}

5. EXAFS analysis did not yield any insights, most likely due to heterogeneity of Co sites in the materials. I praise the authors for being straightforward on it. The literature in this field is littered by structures derived from EXAFS without considering alternatives. Yet to add to my point – due to the complexity and heterogeneity of the material characterization does not deliver any information that cannot be inferred from literature on the topic.

Response: We agree with the reviewer that we could not obtain specific insights from the EXAFS analysis. In our opinion, interpretation of EXAFS fitting is based on obtained available characterization information and providing reasonable hypothesis, but not random attempts. This does not work for Co sites in CoNCB materials due to the complexity and heterogeneity. We believe this is also not feasible for other M-N-C single atom materials which are prepared using annealing strategy. We do not think that this characterization challenge for M-N-C single atom materials has aroused considerable attention. Lots of fancy structures have been still proposed without considering the fundamental knowledge of EXAFS analysis in recent publications (*Nat. Commun.* 2023, 14, 6666; *Nat. Commun.* 2023, 14, 6849; *Nat. Commun.*, 2022, 13, 2963; *Nat. Mater.* 2020, 19, 436–442). Through providing multiple EXAFS fitting alternatives, we appeal to rationally undergo EXAFS analysis and properly elucidate the structure-performance relationship of single atom materials.

6. With rather inconclusive characterization section, the manuscript claim to fame is the excellent performance of the catalysts in hydrogen peroxide synthesis. With so many recent studies reporting “record” performance I with the community could agree on the universal metrics and units before making “highest activity and selectivity” claims. For example, a publication by Cao et al all earlier this year (10.1038/s41467-023-35839-z) claims “record rates” in

similar materials but the units in two papers are different – and Cao’s paper is not cited in the current manuscript. This is a systemic problem, but it makes it difficult to evaluate the paper.

Response: We apologize that we did not demonstrate the performance comparison clearly in the abstract. The highest activity and selectivity of CoNCB was compared to other metal-loading M-N-C materials prepared in our work (MnNCB, FeNCB, NiNCB, and CuNCB), but not to peers’ publications. Although CoNCB sample exhibits excellent performance for electrochemical hydrogen peroxide production, we assume that the performance comparison between different groups is challenging due to various evaluation methods. We appreciate the reviewer agreeing with this systemic problem which is due to non-standardized measurement and experiment procedures. For example, 0.1 M K₂SO₄ (pH 7.2) was used as the neutral electrolyte for RRDE evaluation in the publication by Cao et al. (*Nat. Commun.* 2023, 14, 172 (10.1038/s41467-023-35839-z)) while 0.1 M PBS (pH 7) was used as the neutral electrolyte in our work. In addition, the parameters of gas diffusion electrode setup are also different including the catalyst loading (0.5 mg cm⁻² vs 0.2 mg cm⁻²), electrolyte (0.3 M K₂SO₄ vs 1 M PBS), working area (1 cm² vs 0.25 cm²), the anode material (Pt foil vs nickel foam), and so forth. In this case, we cannot make the proper comparison with Cao’s work. A standard evaluation procedure is required to solve this systemic problem, while this is outside the scope of our present work. We are committed to alleviating this problem through providing comprehensive experimental details in the manuscript.

In addition, the ‘record rate’ proposed by Cao et al. (*Nat. Commun.* 2023, 14, 172) was obtained in alkaline electrolyte which was their focus. We prefer to highlight the significance of electrochemical hydrogen peroxide production in neutral media, which is more feasible for practical applications, as elucidated in the introduction section. *“In contrast to the extensive research of EHPP under alkaline and acid conditions, less efforts have been devoted to the 2e⁻ ORR study in neutral media due to the limitations of ORR kinetics caused by the low concentration of H⁺ and OH⁻ that negatively affects the ORR kinetics.^{10,11} However, the utilization of neutral H₂O₂ solution is more popular and flexible for practical applications, such as household cleaning and gardening, in part as it eliminates the potential secondary cost of neutralizing acid and alkaline solutions.^{8,12} Therefore, it is highly desirable to promote the development of electrocatalysts for EHPP in neutral media, including the design of efficient electrocatalysts, study of novel mechanisms, and endeavor in practical productions.”*

With these deficiencies the publication of the manuscript in Nature Communication is not warranted. My suggestion is to rework the paper and submit it to a more specialized journal.

Response: We appreciate the reviewer’s constructive comments, which motivates us to enhance the

manuscript's quality. The opportunity to address these concerns has allowed us to strengthen our work, offering insights into the rational characterization of single atom catalysts and the significance of electrochemical hydrogen peroxide production in neutral media. We believe these improvements align with the broad audience of Nature Communications. We welcome any further suggestions or comments on our revised manuscript.

Some minor remarks:

Line 162 – perhaps “broadened peaks”

Line 195 – “red shift” is optical spectroscopy jargon. Shift to the lower energy is perhaps more descriptive

Line 202: FT EXAFS is not a “process”

Line 253: replace “inclined” by “increased”

Line 374: I simply do not understand this sentence

Response: We appreciate the reviewer's assistance in correcting these errors. We have revised the manuscript accordingly and highlighted the revision in yellow.

Regarding the sentence in Line 374, ‘perchloric acid (HClO₄), potassium hydroxide (KOH), nafion perfluorinated resin solution (5 wt%)’, we want to clarify that Nafion perfluorinated resin with the concentration of 5 wt% in mixture of lower aliphatic alcohols and water (CAS Number: 31175-20-9) is one of the commercial binders to prepare the catalyst inks. We hope this clarification resolves any confusion.

Reviewer #2 (Remarks to the Author):

The authors reported a series of M-N-C electrocatalysts at ultralow metal-loading for O₂-to-H₂O₂ conversion within neutral electrolyte. Specifically, CoNCB with the asymmetric Co-C/N/O coordinative configuration exhibited the highest electrocatalytic activity and selectivity. To shed light on the structure-performance relationship, comprehensive X-ray radiated characterizations have been carried out, pinpointing the key role of asymmetric Co-C/N/O configuration in promoting H₂O₂ electrosynthesis. To this reviewer, the manuscript is well organized, the electronic structure of prepared catalysts is well characterized by the state-of-the-art synchrotron techniques. I'd suggest its acceptance at Nat. Commun. after addressing the below concerns:

Response: We very much appreciate the reviewer's positive appraisal of our work and featuring our highlights. Point-by-point responses were provided to address the concerns. Further suggestions and comments are very welcome.

1. The authors should double-check the assignment of pyrrolic N peak in N K-edge NEXAFS, as the observed peak position is in controversial to the cited Refs. 36&37.

Response: We greatly appreciate the reviewer pointing out the issue of N K-edge. Unfortunately, we found contamination on N K-edge due to the silicon membrane used in the beamline. N K-edge features were not supposed to exist on the clean bare gold substrate as well (**Figure R2.1**). The peak features at around 398.3 eV, 400.2 eV, 403.0 eV, 406.7 eV and 411.7 eV partly arise from the contamination in the beamline set-up. Silicon window was used to measure this sample at B07-B beamline at Diamond Light Source, in order to avoid the N K-edge interference from silicon nitride window (B07-B beamline setup). The contamination was probably due to the glue used for the silicon window assembly after discussion with beamline scientists. Although the peaks at around 398.3 eV, 401.4 eV, and 406.7 eV features show different features compared to that of bare gold substrate, it is difficult to evaluate the contribution from the contamination in the beamline. We apologize for this as we cannot find a good way to obtain the real N K-edge at the moment. We have added additional supporting notes to the revised supporting information to prevent causing misleading understanding.

Figure R2.1. N K-edge NEXAFS spectra of CoNCB and bare gold substrate. The N K-edge features on bare gold substrate were due to the contamination in the beamline setup.

2. Regarding to the clarification of M-X-C coordination environment within practical single atom catalysts as well as its impact on branching electrocatalytic pathways, I'd direct the authors to some closely related works at [10.31635/ccschem.020.202000667](https://doi.org/10.31635/ccschem.020.202000667) and [10.1038/s41467-022-28346-0](https://doi.org/10.1038/s41467-022-28346-0). Proper remarks should be supplemented.

Response: We appreciate the reviewer providing two excellent references. The authors provided valuable insights on investigating the correlation between Pt-X-C coordination environments and electrocatalytic pathways of practical single atom catalysts. We have cited these two works in our introduction of the revised manuscript. We expect they are spread to more audience in the realm of single atom catalysts and electrocatalysis.

‘Therefore, deeper insights are suggested to understand the electronic structures, such as clarification of coordination environment of M-N-C catalysts, with the aim of revealing the underlying structure-performance relationships.^{26, 27}’

3. In RRDE measurements, the catalyst loading is ca. 4 ug/cm². Proper justification should be provided for this ultra-low catalyst usage, actually it's even hard to fully cover the glassy carbon disk at this dosage.

Response: We understand the reviewer's concern and appreciate the reviewer's suggestion. We investigated the effect of different catalyst loading, including 4 ug/cm², 10 ug/cm², 40 ug/cm², as shown in Figure S15 (first version). They exhibited similar ring current density in the potential range of 0.55-0.75 eV, but the H₂O₂ selectivity significantly decreased for catalyst loading of 40 ug/cm², indicating that the generated peroxide was trapped within catalyst layer and further reduced to H₂O (*Nat. Mater.* 2020, 19, 436–442). Therefore, we opted for a very thin layer of catalyst (4 ug/cm²). In addition, it is worth noting that establishing a standard procedure for catalyst ink preparation is challenging due to variations in materials type, ink ratios and equipment. Particularly for nanomaterials with size ranging from 1-100 nm, they are smaller than the resolution (200 μm) of human eyes and optical microscope (200 nm) used for judging the deposition of ink film. While higher resolution techniques like SEM and AFM offer feasible alternatives, they may not be practical for commercial electrodes. In addition, we believe that it is more critical to keep the same ink preparation procedures since our focus is the electrochemical performance comparison of different M-X-C catalysts. Therefore, we agree with reviewer's suggestion to provide proper justification for the benefit of future researchers and readers. Proper justification was provided in the revised supporting information (under Figure S16).

4. Fig. 3d plots the experimentally determined Tafel slopes for various M-N-C catalysts, where a deepened discussion on the relevant rate-determining step is encouraged, in conjunction with the theoretical simulation part.

Response: We appreciate this valuable suggestion. More Tafel slope analysis has been provided in the revised manuscript. As reported (*J. Electrochem. Soc.* 2012, 159, 11, H864; *Sci. Rep.* 2015, 5, 1, 13801), a Tafel slope close to 120 mV dec⁻¹ indicates that the first charge transfer ($O_2 + * + (H^+ + e^-) \rightarrow OOH^*$) is the rate-determining step. In our work, the Tafel slope of CoNCB is 62 mV dec⁻¹, which is far away from 120 mV dec⁻¹, suggesting the second charge transfer ($OOH^* + (H^+ + e^-) \rightarrow H_2O_2 + *$) is the rate-determining step. This is consistent with our DFT calculation that the binding energy of HOO* is located at the left side of volcano plot, indicating that it is easy to form OOH* but difficult to generate H₂O₂. In addition, it should be noted that the Tafel slope in the manuscript is calculated by the ring current density obtained from the LSV measurement. Tafel slope calculated from disk current density is provided in the revised supporting information (Figure S14), which agrees with the above analysis as well due to the high H₂O₂ selectivity for CoNCB.

Reviewer #4 (Remarks to the Author):

The authors reported on surface-engineered M-N-C, which exhibited high selectivity and activity for the 2e⁻ ORR in a neutral medium. Various characterization methods and DFT simulations were employed to investigate the experimental observations. I found this manuscript to be well-written and interesting. I recommend a major revision.

Response: We sincerely appreciate the reviewer's positive assessment of our manuscript and the insightful suggestions regarding DFT simulations. This constructive feedback inspires us to delve deeper into understanding the structure-performance relationship. The valuable insights gained from these suggestions have been thoughtfully incorporated to further enhance the manuscript.

1. The author stated that "The Gibbs free energy diagram showed the trend of the kinetic barrier at the equilibrium potential of 0.7 V for various models." However, in Figure 5f, it depicts only the energy of the states at 0.7 V, without showing the kinetic barrier. The protonation calculations using explicit water molecules and constant potential calculation scheme are more than 100 times computationally expensive than mere thermodynamic calculations.

Response: We thank the reviewer for correcting the imprecise description in our manuscript. It is crucial to avoid any misunderstanding with the audience. In our study, we used the computational hydrogen electrode model proposed by Nørskov et al (*J. Phys. Chem. B*, 2004, 108, 46, 17886-17892), which does not consider the activation energy of elementary reactions. To accurately reflect this, we have revised the term from 'kinetic barrier' to 'energy barrier' in the manuscript.

2. While Figures 5a and 5b shows H₂O₂ activity and selectivity of CoNCB, respectively, DFT calculation results exclusively highlight the activity for the 2e⁻ ORR. Additional analysis on H₂O₂ selectivity is required.

Response: Thanks for this valuable suggestion. To further elucidate the H₂O₂ selectivity, we have conducted additional analysis based on our calculation results. Generally, the energy difference between H₂O₂ and O* can be used as a descriptor for selectivity for 2e⁻ ORR, where the energy of H₂O₂ is a constant value (*ACS Catal.* 2021, 11, 5, 2483–2491). Selectivity relies on the adsorption strength of O*, where weaker adsorption of O* corresponds to higher selectivity for 2e⁻ ORR. Additionally, a linear scaling relationship exists between OOH* and O*, that is, $E_{\text{OOH}^*} = 0.53E_{\text{O}^*} + 3.18$ (*Chem. Phys.* 2005, 319, 178-184). Accordingly, stronger adsorption of OOH* corresponds to stronger adsorption of O*. Therefore, it can be deduced that weaker adsorption of OOH* is generally better for selective production of H₂O₂. In our research, the adsorption of OOH* was calculated.

Because the energy states of OOH* for all catalyst models are negative, a weaker adsorption OOH* corresponds to a lower energy barrier of protonation step (using a simplified model rather than true kinetic barrier as suggested by the reviewer). Therefore, a weaker adsorption of OOH* is better for both activity and selectivity.

3. The author's explanation for the positive correlation between the off center distance and $G_{d}(\text{OOH}^*)$ is insufficient. At least, an electronic structure analysis is required to provide the underlying reasons.

Response: Thanks for the reviewer's constructive suggestion. Additional electronic structure analysis has been provided. The projected density of states (PDOS) of Co 3d orbitals for three representative models including CoN₄, CoN₄-N, and CoN₄-O models were calculated as displayed in **Figure R3.1a**. Similar electronic structures have been reported but limited to one CoN₄-N or CoN₄-O model (*J. Am. Chem. Soc.* 2018, 140, 12, 4218–4221; *Energy Environ. Sci.*, 2021, 14, 5444–5456; *Adv. Funct. Mater.*, 2022, 32, 49, 2209499). Higher occupation of Co 3d states is observed for CoN₄ near the Fermi level, indicating stronger adsorption for intermediates. In contrast, CoN₄-O and CoN₄-N exhibit decreased occupation states and near the Fermi level broadened peaks, suggesting weaker adsorption strength due to ligand-induced electron delocalization. Furthermore, the d-band centers were calculated as descriptors for adsorption strength, with more negative values indicating less active adsorption sites. Both CoN₄-O and CoN₄-N show more negative d-band center compared to CoN₄, with CoN₄-N exhibiting the most negative center due to the stronger interaction between Co and N, which is because the electronegativity of N is stronger than O. The charge density difference of CoN₄-N and CoN₄-O models are shown in **Figure R3.1b** with an isosurface level of 0.01 e/bohr³. There is a charge redistribution after the conformation of Co-N and Co-O bonds, where the charge is accumulated. A stronger charge redistribution appears for CoN₄-N compared with CoN₄-O, indicating a stronger coordination bond of Co-N than Co-O. On the opposite position of Co-N and Co-O bonds, the charge density decreases, thus making the adsorption strength weaker. Nevertheless, other factors, such as the additional carbon layer and bond length, probably contribute to enhanced adsorption energy as well. The effect of additional carbon layer is further discussed in response to comment 5.

Figure R3.1. (a) The projected density of states (PDOS) of Co 3d orbitals of CoN₄ models with different ligands. Charge density difference of (b) CoN₄-N and (c) CoN₄-O models. The isosurface level is 0.01 e/bohr³. The yellow and blue region denotes accumulation and decrease of electron density respectively.

4. Despite of highlighting the reaction in a neutral media, the DFT calculations were performed under the vacuum system. Recently, the constant potential calculation using implicit solvation method is becoming a standard in the field. Since this method can include the effect of pH, cation, etc, it is highly recommended to perform this method.

Response: We appreciate your suggestion, and as recommended, we have re-calculated all the adsorption models using the implicit solvation method with the VASPsol code (*J. Chem. Phys.*, 2019, 151, 234101; *J. Chem. Phys.*, 2014, 140, 084106). The inclusion of the implicit water environment resulted in a shift of about 0.17-0.27 eV for G_{OOH*}. While the adsorption strength of OOH on all catalyst models was overestimated in calculations without the solvent method, this adjustment does not alter our previous conclusions. The new data has been incorporated into the manuscript to enhance accuracy and precision.

Table 3.1. The difference between the calculated G_{OOH*} with and without using implicit solvation method.

Model	G _{OOH*} with solvent model (eV)	G _{OOH*} without solvent model (eV)	Difference (eV)
CoN ₂ C ₂ -C	3.75	3.94	0.19

CoN ₂ O ₂ -C	3.66	3.86	0.20
CoN ₄ -C	3.63	3.90	0.27
CoN ₃ C ₁ -C	3.48	3.73	0.25
CoN ₄ -N	3.58	3.82	0.24
CoN ₄ -O	3.43	3.67	0.23
CoN ₄	3.23	3.50	0.27
CoN ₃ O ₁ -O	3.22	3.39	0.17
CoN ₂ O ₂ -O	3.21	3.43	0.22

5. The configurations of CoNCB were modelled with an additional carbon layer for axial ligand. Is the enhanced performance result from an axial ligand or an additional carbon layer or both? The merit of DFT calculations is that several effects can be separated by properly modelling the simulations.

Response: We appreciate this insightful suggestion. While the impact of axial ligands in M-N-C configurations has been extensively studied, the role of an additional carbon layer has received limited attention (*J. Am. Chem. Soc.* 2018, 140, 12, 4218–4221; *Energy Environ. Sci.*, 2021, 14, 5444-5456; *Adv. Mater.*, 2022, 34, 29, 2202544; *Angew. Chem., Int. Ed. Engl.*, 2023: e202304625). To assess the effect of additional carbon layer, we calculated CoN₄/N model and CoN₄/O model which incorporate CoN₄ with additional N- and O- doped carbon layer, respectively (**Figure R3.2**). A hydrogen atom was used for the saturation of the active N or O ligand. The G_{ad}(OOH*) of CoN₄/N model and CoN₄/O model is 3.16 eV and 3.25 eV, respectively, which is close to that of CoN₄ model (3.23 eV). In contrast, CoN₄-N and CoN₄-O with axial ligand, exhibit G_{ad}(OOH*) values of 3.58 eV and 3.38 eV, respectively, suggesting that axial ligand critically influences the G_{ad}(OOH*). Nevertheless, evaluating the role of the additional carbon layer in adjusting the bond length of the axial ligand and the off-center distance of the central Co atom is challenging. The model system is intricately balanced by coordinate bonds and Van der Waals interactions between the carbon layers. Therefore, we assume that the enhanced performance likely results from the synergetic effect of off-center distance, coordination atoms and length, and layer distance. While it is challenging to isolate the individual roles, we have provided this analysis in the revised manuscript and supporting information, aiming to inspire future research in the field of single atom catalysts.

Figure R3.2 The models and corresponding adsorption energies without a carbon layer and a ligand (left), with a carbon layer but without a ligand (middle) and with both a carbon layer and a ligand (right).

REVIEWER COMMENTS

Reviewer #2 (Remarks to the Author):

The authors have properly addressed my previous concerns, it could be published in the current form.

Reviewer #4 (Remarks to the Author):

- The authors did not compare their results with literature. In Nature Materials volume 19, pages 436–442 (2020), Figure 1A says $G(\text{OOH}^*) \sim 3.9$ eV for Co-N₄, but the Figure 5g in this draft says $G(\text{OOH}^*) \sim 3.2$ eV. Other works also mention ~ 4.0 eV of $G(\text{OOH}^*)$. Without a clarification of this result, further results and discussions are doubtful.

- The author mentioned that the linear scaling relationship between OOH^* and O^* enables us to deduce O^* through OOH^* , indicating that the adsorption tendency of OOH could imply the selective production of H_2O_2 . However, the specific value of O can be determined through DFT calculations. I recommend that after obtaining the O^* value through calculations, comparing the selectivity with the energy of H_2O_2 and O^* would offer a more intuitive understanding of the selectivity.

- The author established a correlation between the d-band center and $G(\text{OOH}^*)$ in three models for CON₄, CON₄-N, and CON₄-O. What do the electronic structure analysis results indicate when C ligands are incorporated into CON₄ in addition to O and N ligands? Furthermore, the nine models depicted in Figure 5g, what are the outcomes for the remaining six models not discussed?

- From a theoretical calculation perspective, given that MnN₄ demonstrates a stronger adsorption energy compared to CoN₄, implying its position further to the left on the volcano plot, it would be expected that the rate-determining step for MnN₄ is the second charge transfer ($\text{OOH}^* + (\text{H}^+ + \text{e}^-) \rightarrow \text{H}_2\text{O}_2 + *$). However, Figure 3d reveals that the Tafel slope for MnN₄ is ~ 120 mVdec⁻¹, suggesting that the rate-determining step for MnN₄ is the first charge transfer ($\text{O}_2 + * + (\text{H}^+ + \text{e}^-) \rightarrow \text{OOH}^*$). This raises the question of whether the experimental results for MnNCB align with the DFT calculation predictions regarding the rate-determining step.

Reviewer #5 (Remarks to the Author):

The authors (NCOMMS-23-26996A) reported a series of M-N-C single-atom catalysts by surface-engineering method. The Co-N-C catalyst exhibited high selectivity and activity in the 2e⁻ ORR under a neutral medium. Various characterization methods were carried out and DFT simulations were further employed to investigate structure-performance relationships.

Generally, this work is solid and provides some novel insights into the electronic configurations of electrocatalysts. The authors have adequately addressed the previous comments and it should be acceptable for Nat. Commun. after addressing the following issues.

1. As Reviewer 1 pointed out (comment 1), the HAADF-STEM images are poor evidence to claim the Co single atoms, the images taken under the dark field for the catalyst at low resolution (e.g. 50 nm, 100 nm scale bar) would be helpful to exclude the Co nanoparticles.
2. Introduction part: the authors correctly mentioned, "However, the limitations of fitting FT-EXAFS have been sometimes arbitrarily neglected on distinguishing M-C/N/O coordination with similar bond lengths due to close atomic number, thus leading to ambiguous or even somewhat misleading". However, the current XAFS data analysis only provides a vague asymmetric structure. The M-C/N/O coordination was not distinguished by EXAFS fitting in this work, and the basis for the improvement in

catalytic performance can only be inferred from DFT, rendering the paper's conclusion on the structure-activity relationship insufficient.

3. Please correct the caption for Figure 2, regarding the Co L-edge (which has been removed from Figure 2).

4. Please modify the color of hydrogen in the legend in Figure 1c, it is difficult to read. The proposed trend of activity under the reaction condition (acidic, neutral, or alkaline) should be marked.

5. Page 13 (difficult to refer to without page or line #!), below the caption of Figure 5, The X-ray spectroscopy analysis of CoNCB material, what spectroscopy?

6. XANES simulation would be more structure-sensitive compared to EXAFS, which may be helpful in figuring out the structure of M-N-C.

7. What are the reasons for the formation of asymmetric Co-C/N/O configurations on CoNCB by the synthesis method? As normally Metal-N₄ is dominant among reported references.

8. The authors claim that the CoNCB catalyst exhibits the best performance, especially in terms of the catalytic activity of Co (6.1×10^5 A gCo⁻¹). However, the NCB catalyst also showed some activity, indicating that the performance of CoNCB may not solely be attributed to the Co sites, but could also be partially influenced by the contribution of NCB supports?

Reviewer 2

The authors have properly addressed my previous concerns, it could be published in the current form.

Response: We appreciate the reviewer's positive feedback and thank the help for improving our manuscript.

Reviewer #4

- The authors did not compare their results with literature. In *Nature Materials* volume 19, pages 436–442 (2020), Figure 1A says $G(\text{OOH}^*) \sim 3.9$ eV for Co-N₄, but the Figure 5g in this draft says $G(\text{OOH}^*) \sim 3.2$ eV. Other works also mention ~ 4.0 eV of $G(\text{OOH}^*)$. Without a clarification of this result, further results and discussions are doubtful.

Response: We sincerely apologize for not comparing our results with the literature and highly appreciate you pointing this out. After thoroughly re-evaluating our data, we realized that we forgot to add the corrections of ZPE and TS items (although they were mentioned in the method section). After incorporating these corrections, the $G_{\text{ad}}(\text{OOH}^*)$ for CoN₄ is 3.63 eV. Nevertheless, we acknowledge that there is still a residual difference of 0.27 eV from literature value (*Nature Materials* volume 19, pages 436–442 (2020)). We believe that this discrepancy arises from the variations in calculation methods, such as dispersion corrections and solvation adjustments. In addition, diverse models also cause different outcomes. For example, the adsorption energy of models would be impacted by the supercell size and lattice constants. It is worth noting that after revision, the overall trend remains consistent with most data points aligning with the left side of the volcano curve, except for CoN₂O₂-C. We sincerely apologize for the oversight in not including ZPE and TS corrections in the previous version. We believe these corrections and explanation would strengthen the accuracy of our results.

- The author mentioned that the linear scaling relationship between OOH^* and O^* enables us to deduce O^* through OOH^* , indicating that the adsorption tendency of OOH could imply the selective production of H_2O_2 . However, the specific value of O can be determined through DFT calculations. I recommend that after obtaining the O^* value through calculations, comparing the selectivity with the energy of H_2O_2 and O^* would offer a more intuitive understanding of the selectivity.

Response: We appreciate your suggestion. As suggested, we conducted additional

calculations to determine the adsorption of O^* . Prior to this, we revised the models with axial carbon ligand including CoN_4-C , CoN_3C_1-C , CoN_2C_2-C , and CoN_2O_2-C . This was because we found that, in the previous model (Figure R2.1a), the adsorption of O^* on the Co atom led to the breakage of the axial ligand (fifth ligand). To ensure a more stable catalyst, we introduced a new model by removing an additional C atom from the bottom carbon layer. In this new model, the catalyst's structure remains unchanged upon O^* adsorption. We acknowledge the practical existence of various defects on carbon layers but characterizing them precisely is challenging. While the old structure may exist in our samples, the use of a more stable catalyst model facilitates a clearer discussion of the scientific aspects.

Our results support our previous conclusions. Figure R2.1b displays the relationship of the calculated $G_{ad}(OOH^*)$ and $G_{ad}(O^*)$. It indicates that a weaker adsorption of OOH^* corresponds to a weaker adsorption of O^* . As a descriptor of 2e ORR selectivity (*ACS Catal.* 2021, 11, 5, 2483–2491), $G_{ad}(H_2O_2) - G_{ad}(O^*)$ will be more negative with a weaker adsorption of OOH^* , where $G_{ad}(H_2O_2)$ is a constant. According to our approximation, a more positive $G_{ad}(OOH^*)$ is more likely to exhibit higher selectivity for H_2O_2 in the context of the descriptor $G_{ad}(H_2O_2) - G_{ad}(O^*)$, where $G_{ad}(H_2O_2)$ is a constant. It's worth noting that our linear relationship differs from the literature (*Chemical Physics* 2005, 319, 178-184), where $E_{OOH^*} = 0.53E_{O^*} + 3.18$. This discrepancy can be attributed to the use of metal models in the literature, whereas our study focuses on single-atom catalyst models.

Figure R2.1. (a) Previous and current carbon layer structure used for configurations with axial carbon. (b) The relationship between calculated $G_{ad}(OOH^*)$ and $G_{ad}(O^*)$. (c) The symbols of different configurations.

- The author established a correlation between the d-band center and $G_{ad}(OOH^*)$ in three models for CoN_4 , CoN_4-N , and CoN_4-O . What do the electronic structure analysis results indicate when C ligands are incorporated into CoN_4 in addition to O and N ligands? Furthermore, the nine models depicted in Figure 5g, what are the outcomes for the remaining six models not discussed?

Response: We apologize that we did not provide all models in the first round. The density of states and electron density difference for all models are provided in Figure R2.2. The relationship among d-band center, $G_{ad}(OOH^*)$ and $G_{ad}(O^*)$ aligns with literatures, where more negative d-band center corresponds to weaker adsorption of intermediates (Figure R2.2a) (Surf. Sci. 1995, 343, 3, 211-220). Interestingly, a positive relationship between Co off-center distance and negative d-band center shift was found, indicating that more distorted configuration presents weaker adsorption of intermediates (Figure R2.2b). Of note, CoN_2O_2-C configuration with the largest off-center distance is an outlier. It is probably because the in-plane O is saturated by surrounding C atoms, leading to the weakened interaction between Co and O ligand. Moreover, the interaction between Co and axial C ligand is very active, thus pulling the Co far away from the plane center compared to that of axial N and O ligand.

Figure R2.2. (a) The relationship between $G_{ad}(OOH^*)$, $G_{ad}(O^*)$ and d-band center of different models. (b) The projected density of states (PDOS) of different models. (c) The relationship between off-center distance and d-band center of different models. (d) Charge density difference of different models. The iso-surface level is $0.01 e/bohr^3$. The yellow and blue region denotes accumulation and decrease of electron density respectively. These figures have been added into the revised manuscript (Figure 5) and supporting information (Figure S24).

- From a theoretical calculation perspective, given that MnN_4

demonstrates a stronger adsorption energy compared to CoN₄, implying its position further to the left on the volcano plot, it would be expected that the rate-determining step for MnN₄ is the second charge transfer ($\text{OOH}^* + (\text{H}^+ + \text{e}^-) \rightarrow \text{H}_2\text{O}_2 + *$). However, Figure 3d reveals that the Tafel slope for MnN₄ is $\sim 120 \text{ mVdec}^{-1}$, suggesting that the rate-determining step for MnN₄ is the first charge transfer ($\text{O}_2 + * + (\text{H}^+ + \text{e}^-) \rightarrow \text{OOH}^*$). This raises the question of whether the experimental results for MnNCB align with the DFT calculation predictions regarding the rate-determining step.

Response: We understand your concern and appreciate you pointing this out. It has been generally reported that MnN₄ configuration possesses stronger adsorption energy compared to FeN₄ and CoN₄ configuration from DFT perspective, which is supposed to be very active for ORR (*Chem*, 2020, 6, 658-674; *J. Am. Chem. Soc.*, 2019, 141, 12372–12381; *Nat. Catal.*, 2018, 1, 5, 339-348; *Angew. Chem., Int. Ed. Engl.*, 2016, 55, 14510-14527). However, from experimental perspective, the Mn-N-C single atom catalyst for ORR has been barely reported compared to Fe- and Co- based single atom catalysts. Gao et al. proposed that Mn sites would be blocked due to its too strong adsorption of intermediates, thus exhibiting similar electrochemical behaviors compared to bare metal-free NC materials (*Chem*, 2020, 6, 658-674). This makes it struggle to well align the DFT calculations predictions with the experimental results for Mn-N-C material. In our work, MnNCB material displayed similar Tafel slope with NCB, which could also be probably due to the ultra-low metal loading, thus making it more complicated to align the DFT calculations with experimental results for MnNCB material. To avoid misunderstanding, the related description has been revised as follows:

*“In addition, the Tafel slopes determined by the LSV current density displayed similar behaviors (Figure S15), exhibiting the fast kinetic activity for CoNCB material, compared to other prepared M-N-C electrocatalysts. The Tafel slope of CoNCB is far away from 120 mV dec^{-1} , probably indicating that the first charge transfer ($\text{O}_2 + * + (\text{H}^+ + \text{e}^-) \rightarrow \text{OOH}^*$) is not the rate-determining step.⁴⁶ Of note, the analysis of Tafel slopes of other M-N-C materials exhibited discrepancies with literatures^{3, 45}, which might be due to the different metal loadings in the electrocatalysts.”*

Reviewer 5

The authors (NCOMMS-23-26996A) reported a series of M-N-C single-atom catalysts by surface-engineering method. The Co-N-C catalyst exhibited high selectivity and activity in the 2e ORR under a

neutral medium. Various characterization methods were carried out and DFT simulations were further employed to investigate structure-performance relationships.

Generally, this work is solid and provides some novel insights into the electronic configurations of electrocatalysts. The authors have adequately addressed the previous comments and it should be acceptable for Nat. Commun. after addressing the following issues.

Response: We appreciate the reviewer's positive appraisal of our work, along with highly valuable comments and suggestions. Point-by-point responses were provided to address the following comments.

1. As Reviewer 1 pointed out (comment 1), the HAADF-STEM images are poor evidence to claim the Co single atoms, the images taken under the dark field for the catalyst at low resolution (e.g. 50 nm, 100 nm scale bar) would be helpful to exclude the Co nanoparticles.

Response: We appreciate this valuable suggestion. STEM images of CoNCB material with both dark field and bright modes at lower magnification (60k, scale bar:50nm) were provided in the revised supporting information (Figure R2.3). As mentioned in the response to reviewer 1 in the first-round revision, the dense carbon black lattice with stronger scattering ability exhibits brighter signal in the HAADF-STEM image and darker signal in the ABF-STEM image, compared with the amorphous carbon area. Nevertheless, if Co nanoparticles exist, a more obvious contrast was supposed to be present due to the higher scattering ability of Co nanoparticles lattice. However, this was not observed in CoNCB. Therefore, we believe that these additional STEM images further confirm the atomic dispersion of Co species, combining with the characterization of high magnification of HAADF-STEM image with angstrom resolution and Co K-edge XAFS analysis.

Figure R2.3. (a) HAADF-STEM image and (b) ABF-STEM image of CoNCB. The magnification is 60k and scale bar is 50 nm. This figure has been added to the revised supporting information (Figure S2).

2. Introduction part: the authors correctly mentioned, “However, the limitations of fitting FT-EXAFS have been sometimes arbitrarily neglected on distinguishing M-C/N/O coordination with similar bond lengths due to close atomic number, thus leading to ambiguous or even somewhat misleading”.

However, the current XAFS data analysis only provides a vague asymmetric structure. The M-C/N/O coordination was not distinguished by EXAFS fitting in this work, and the basis for the improvement in catalytic performance can only be inferred from DFT, rendering the paper's conclusion on the structure-activity relationship insufficient.

Response: We appreciate your positive appraisal of our description regarding the limitations of EXAFS fitting in the introduction. Meanwhile, we understand your concern regarding not distinguishing M-C/N/O by EXAFS fitting. We provided three fitting results in Figure S12-14 and Table S4 to show the audience that there were many possibilities for EXAFS fitting. This indicates that EXAFS fitting is not a good way to interpret the accurate M-C/N/O coordination for carbonized M-N-C materials, in particular, considering the complexity and heterogeneity of various M-C/N/O configurations in M-N-C materials. We believe that this is a crucial challenge in the research of M-N-C single atom catalyst materials. Nevertheless, based on the XANES analysis and structure analysis of our materials, we could deduce the distorted and

asymmetric Co-C/N/O configurations. Moreover, XANES simulation was provided thanks to your suggestion (Figure R2.5). We also used the commercial cobalt porphyrin molecule with known CoN_4 to compare its electrochemical performance with CoNCB material. Furthermore, different from previous studies which focused on the individual asymmetric configuration, we provided the quantitative evaluation of the relationship of distorted degree of asymmetric configurations and their activity based on the DFT study. We believe that this work provided novel insights for understanding the structure characterization and structure-activity relationship in the research of M-N-C single atom catalysts.

3. Please correct the caption for Figure 2, regarding the Co L-edge (which has been removed from Figure 2).

Response: We apologize for this mistake. The wrong caption has been removed from the revised manuscript.

4. Please modify the color of hydrogen in the legend in Figure 1c, it is difficult to read. The proposed trend of activity under the reaction condition (acidic, neutral, or alkaline) should be marked.

Response: We appreciate your valuable suggestion. The color of hydrogen in the legend of Figure 1c has been modified to a darker color to enhance readability. Additionally, the reaction condition has been added in caption in the revised manuscript.

Figure R2.4. The schematic illustration of the EHPP selectivity and activity of centrosymmetric CoN_4 and asymmetric Co-C/N/O electrocatalysts in 0.1 M PBS (pH = 7). This figure has been updated in the revised manuscript (Figure 1c).

5. Page 13 (difficult to refer to without page or line #!), below the

caption of Figure 5, The X-ray spectroscopy analysis of CoNCB material, what spectroscopy?

Response: We apologize for the mistake. It is the Co K-edge XAFS spectroscopy analysis. Correct description has been added into the revised manuscript.

6. XANES simulation would be more structure-sensitive compared to EXAFS, which may be helpful in figuring out the structure of M-N-C.

Response: We appreciate this valuable suggestion. XANES simulation using Finite Difference Method Near Edge Structure (FDMNES) program was performed based on the Crystallographic Information File (CIF) of different DFT-optimized configurations. The simulated XANES spectra of Co foil and CoPr standard align well with the experimental XANES spectra. Nevertheless, it should be noted that simulated XANES spectra display more significant electron-transition features than the experimental ones. For different Co-C/N/O configurations, the feature of dipole forbidden 1s to 3d electron transition inclines due to the 3d and 4p orbital hybridization in the asymmetric configurations. Moreover, along with the increase of distortion degree (off-center distance), the intensity of this feature increases. Besides, the 1s-4p_z transition and ligand-to-metal charge transfer shakedown feature was absent for all asymmetric configurations. Nevertheless, we assume that it is still difficult to figure out the accurate structure of M-N-C due to the complexity and heterogeneity of various M-C/N/O configurations in M-N-C materials. Therefore, we had to employ DFT to study the correlation between different Co-C/N/O configurations and their 2e⁻ ORR activity.

Figure R2.5. (a) Experimental and simulated XANES spectra of Co foil and CoPr. (b) Simulated XANES spectra of different configurations. These figures have been added into the revised manuscript (Figure 5) and supporting information (Figure S23).

7. What are the reasons for the formation of asymmetric Co-C/N/O configurations on CoNCB by the synthesis method? As normally Metal-N₄ is dominant among reported references.

Response: We appreciate this insightful question. We believe that the main reason arises from the unique structure of carbon black materials. Ketjenblack EC-300J carbon black (AkzoNobel Co., Ltd.) was used in our synthesis method to support Co species. The carbon black support exhibited compact shell consisting of graphitic layers. The outside surface was distorted with lots of defects that were induced by HNO₃ treatment, as shown in Figure 2 (in the manuscript). The compact graphitic shell prevented burring metal atoms, while the distorted outside surface with defects caused the formation of symmetric Co-C/N/O configurations, thus exposing substantial active sites. We agree that metal-N₄ is normally dominant among previous reported references, which was probably inspired by the metal phthalocyanine, porphyrins or other derivatives which possess known metal-N₄ configurations. (*Nature*, 1964, 201, 1212–1213; *J. Phys. Chem.*, 1992, 96, 10898-10905; *Chem. Rev.*, 1988, 88, 1121-1146; *Russ. J. Electrochem.*, 2004, 40, 1174-1187). On the other hand, asymmetric configurations have been widely reported recently. Nevertheless, more in-depth studies are required without limiting to the individual asymmetric configuration after acquiring the limitations of EXAFS analysis.

We have added this description in our revised manuscript:

“In addition, as displayed in the STEM images, the unique and distorted surface of carbon black support probably contributed to the formation of asymmetric configurations.”

8. The authors claim that the CoNCB catalyst exhibits the best performance, especially in terms of the catalytic activity of Co (6.1×10⁵ A gCo⁻¹). However, the NCB catalyst also showed some activity, indicating that the performance of CoNCB may not solely be attributed to the Co sites, but could also be partially influenced by the contribution of NCB supports?

Response: We appreciate your inspiring comment. The contribution of NCB support is critical to the excellent performance of CoNCB. First, NCB catalyst also showed activity and especially high selectivity (0.2 – 0.4 V vs. RHE). Besides, NCB played a crucial role for anchoring atomical dispersion of metal-C/N/O species with distorted and asymmetric configurations (as mentioned in the response to the above comment). Nevertheless, it is worth noting the indispensable role of Co-C/N/O species in

enhancing the onset potential of electrochemical hydrogen peroxide production. As shown in Figure 3a, the onset potential (potential at 0.1 mA cm^{-2}) of CoNCB is 260 mV more positive than metal-free NCB. The high TOF value was determined at 0.5 V vs. RHE, where the current density of CoNCB is 2.56 mA cm^{-2} , while it is only 0.1 mA cm^{-2} for NCB.

We sincerely appreciate all reviewers' time and dedication for further strengthening our manuscript for *Nature Communications* and welcome additional comments and suggestions.

REVIEWERS' COMMENTS

Reviewer #4 (Remarks to the Author):

The authors addressed the comments well. The followings are some points requiring a clarification.

1. The author stated that CoN2O2-C, identified as an outlier due to its largest off-center distance, exhibits in-plane O saturation by surrounding C atoms, resulting in a weakened interaction between Co and the O ligand. However, in the case of CoN2O2-O, where in-plane O is also saturated by surrounding C atoms, a different trend is observed. Therefore, could the author employ COHP to elucidate the weakened interaction between Co and the O ligand, providing a comprehensive analysis of the bond interactions between Co and O?

2. The Y-axis label in Figure S24.a should be corrected from "Gad(OH*)" to "Gad(O*)".

Reviewer #5 (Remarks to the Author):

The Authors have addressed my comments/concerns and I recommend the acceptance of this manuscript.